# A quantitative in vivo CRISPR-imaging platform identifies regulators of hyperplastic and hypertrophic adipose morphology in zebrafish

**Rebecca Wafer†, Panna Tandon†, James Minchin***

Institute for Neuroscience and Cardiovascular Research, Edinburgh BioQuarter, College of Medicine and Veterinary Medicine, University of Edinburgh, Edinburgh, United Kingdom

## eLife Assessment

In this manuscript, Wafer and Tandon et al. present a thoughtful and well-designed genetic screen for regulators of adipose remodeling using zebrafish as a model system. This work is **valuable** because it uncovers several genes associated with adipose tissue hyperplastic hypertrophic morphology and diet-induced remodelingthe hat have considerable potential health impact. The rigorous phenotypic analyses and **compelling** evidence make this work a key resource for the field.

## Abstract

Adipose tissues exhibit a remarkable capacity to expand, regress, and remodel in response to energy status. The cellular mechanisms underlying adipose remodelling are central to metabolic health. Hypertrophic remodelling – characterised by the enlargement of existing adipocytes – is associated with insulin resistance, type 2 diabetes, and cardiovascular disease. In contrast, hyperplastic remodelling – in which new adipocytes are generated – is linked to improved metabolic outcomes. Despite its clinical importance, the regulation of hypertrophic and hyperplastic adipose morphology remains poorly understood. Here, we integrate human transcriptomic data with a quantitative CRISPR-imaging platform in zebrafish to identify regulators of adipose morphology. We developed an image-based phenotyping pipeline that captures lipid droplet size, number, and spatial patterning, and applied generalised additive modelling to quantify hyperplastic versus hypertrophic morphology signatures. Using this platform, we conducted an F0 CRISPR screen targeting 25 candidate genes and identified three that induced hypertrophic morphology (*txnipa*, *mmp14b*, and *foxp1b*) and an additional candidate that altered total adiposity (*kazna*). For functional validation, we generated stable loss-of-function alleles for both zebrafish foxp1 paralogues. Spatial analysis along the anterior-posterior axis revealed that *foxp1b* mutants display developmental hypertrophy but profoundly blunted adaptive responses to high-fat diet (~68% reduction across all spatial zones), while *foxp1a* mutants show normal baseline morphology but disrupted spatial patterning of diet-induced hypertrophy. Together, these findings establish a scalable CRISPR-imaging platform for in vivo genetic screening of adipose morphology and reveal distinct roles for Foxp1 paralogues in developmental patterning and adaptive responses to dietary challenge in adipose tissue.

## Introduction

Adipose tissues exhibit a remarkable capacity to remodel and restructure themselves in response to the energetic status, the environment, physiological demands, or disease. Adipose remodelling

*For correspondence:
james.minchin@ed.ac.uk

†These authors contributed equally to this work

Competing interest: The authors declare that no competing interests exist.

involves alterations in both adipocyte number (addition or removal of adipocytes) and adipocyte size (expansion or reduction in adipocyte size). Increases in adipocyte size (hypertrophy) can trigger hypoxia, necrosis, altered lipid flux, inflammation, and insulin resistance (*Hagberg and Spalding, 2024*; *Tandon et al., 2018*; *Vishvanath and Gupta, 2019*). Accordingly, a hypertrophic adipose morphology – adipose characterised by few, large adipocytes – is associated with increased cardiometabolic disease (*Hagberg and Spalding, 2024*; *Rydén and Arner, 2017*; *Weyer et al., 2000*). Contrastingly, a hyperplastic adipose morphology – characterised by many small adipocytes – is associated with more metabolically beneficial outcomes (*Boden et al., 2003*; *Hoffstedt et al., 2010*). Indeed, reductions in the hyperplastic potential of mouse adipose are associated with decreased insulin sensitivity (*Kim et al., 2014*), and disruption of adipocyte progenitor differentiation, proliferation, and renewal leads to adipose hypertrophy and ensuing metabolic dysfunction (*Gao et al., 2020*). Strikingly, adipose hypertrophy is increased and preadipocyte frequency is reduced in diabetic obese patients (relative to non-diabetic obese) (*Muir et al., 2016*), and genetic predisposition for type 2 diabetes, but not obesity, is associated with an impaired ability to recruit new adipose cells to store excess lipids in the subcutaneous adipose tissue (*Arner et al., 2011*). Despite its clinical relevance, the genetic and cellular processes that determine hyperplastic versus hypertrophic adipose morphology remain poorly understood, and whether early developmental patterning of adipose morphology influences subsequent capacity for adaptive remodelling remains an open question.

The zebrafish provides a tractable and powerful in vivo model for studying adipose growth and remodelling dynamics. Zebrafish adipose tissue is morphologically similar to mammalian white adipose tissue (WAT), comprising large adipocytes with a single dominant cytoplasmic lipid droplet (LD) (*Flynn et al., 2009*; *Imrie and Sadler, 2010*). Transcriptomic profiling further supports this conservation, with RNA-Seq data revealing strong molecular similarity between zebrafish and mammalian WAT (*Morocho-Jaramillo et al., 2024*). Functionally, zebrafish adipose depots respond dynamically to nutritional status – expanding following a high-fat diet (HFD) and regressing following food restriction (*Minchin et al., 2018*; *Minchin and Rawls, 2017*). Further, prolonged high-fat or overfeeding has been shown to induce diabetic phenotypes in zebrafish (*Landgraf et al., 2017*; *Oka et al., 2010*), reinforcing the model's relevance to metabolic disease. One of the key advantages of the zebrafish model is its optical accessibility and amenability to fluorescent imaging. Lipophilic dyes such as Nile Red can be used to label LDs in vivo (*Flynn et al., 2009*; *Minchin and Rawls, 2011*), and LD-associated protein reporters now allow real-time visualisation of LD dynamics in living animals, enabling rapid assessment of adiposity and LD morphology (*Lumaquin et al., 2021*; *Wilson et al., 2021*). Additionally, transgenic tools are increasingly available to label and manipulate zebrafish adipocytes (*Lepanto et al., 2021*; *Mao et al., 2021*). Importantly, zebrafish are amenable to high-throughput CRISPR mutagenesis, and recent studies have combined F0 screens with quantitative imaging to examine candidate obesity genes (*Mazzaferro et al., 2025*; *van der Klaauw et al., 2019*). However, existing approaches have focused on total adiposity rather than cellular composition of adipose tissue. To date, no scalable platform exists to quantify hyperplastic versus hypertrophic adipose morphology in vivo – a gap that limits systematic genetic dissection of adipose growth properties.

Here, we integrate human transcriptomic data with a quantitative CRISPR-imaging platform in zebrafish to identify regulators of adipose morphology. We developed an image-based phenotyping pipeline that captures LD size, number, and spatial patterning within subcutaneous adipose, and applied generalised additive modelling to quantify hyperplastic versus hypertrophic morphology signatures. Using this platform, we screened 25 candidate genes and identified three that induced hypertrophic morphology (*txnipa*, *mmp14b,* and *foxp1b*) and one that increased total adiposity (*kazna*). For functional validation, we focused on Foxp1, a transcription factor known to regulate stem and progenitor cell maintenance across multiple tissues. We generated stable loss-of-function alleles for both zebrafish *foxp1* paralogues and performed spatial analysis of LD dynamics along the anterior-posterior axis. This revealed that *foxp1b* mutants display developmental hypertrophy but show profoundly blunted adaptive responses to HFD, while *foxp1a* mutants show normal baseline morphology but altered spatial patterning of diet-induced hypertrophy. These findings suggest distinct roles for Foxp1 paralogues in developmental patterning versus adaptive remodelling of adipose tissue.

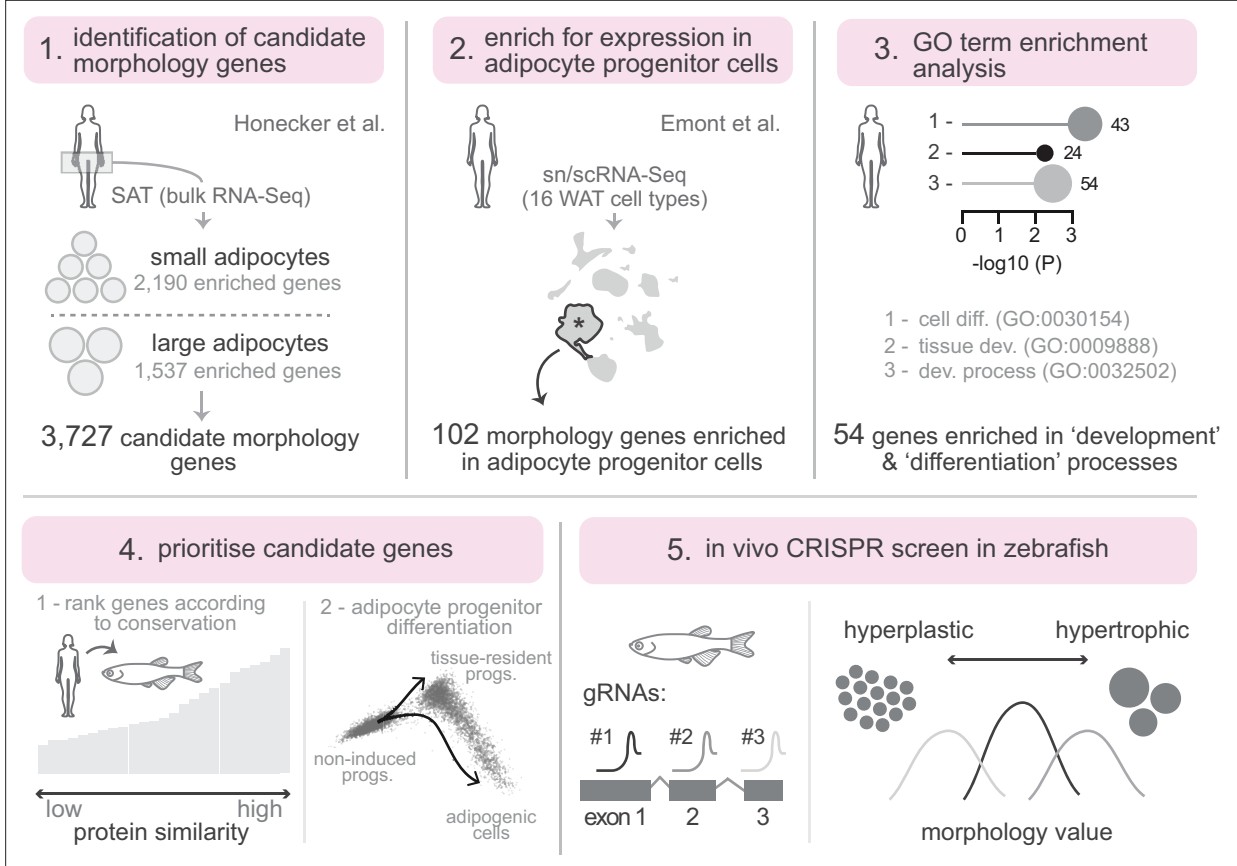

**Figure 1.** Experimental workflow. (1) Identification of candidate morphology genes from bulk RNA-Seq of human subcutaneous adipose tissue (SAT), comparing genes enriched in small versus large adipocytes (Honecker et al.). (2) Filtering for expression in adipose stem and progenitor cells using sn/scRNA-Seq of 16 white adipose tissue (WAT) cell types (Emont et al.). (3) Gene Ontology (GO) term enrichment analysis to focus on genes involved in development and differentiation processes. (4) Prioritisation of candidate genes based on protein conservation to zebrafish and expression dynamics during adipocyte progenitor differentiation. (5) In vivo F0 CRISPR screen in zebrafish using multiple guide RNAs (gRNAs) per gene to quantify lipid droplet morphology (hyperplastic versus hypertrophic).

The online version of this article includes the following figure supplement(s) for figure 1:

**Figure supplement 1.** Expression clustering of candidate adipose morphology genes across human white adipose tissue (WAT) cell types.

**Figure supplement 2.** Integrative analysis of adipose morphology genes across cardiometabolic traits, genome-wide association study (GWAS) loci, Gene Ontology (GO) terms, and single-cell expression profiles.

## Results

### Identification of 102 candidate human adipose morphology genes enriched in adipocyte stem and progenitor cells

To identify candidate regulators of hyperplastic/hypertrophic adipose morphology, we leveraged previously published bulk RNA-Seq data that reported 3727 differentially expressed genes (DEGs) in human subcutaneous adipose tissue (SAT) characterised by either large or small adipocytes (*Figure 1*; *Supplementary file 1*; *Honecker et al., 2022*). We reasoned these DEGs may contribute to defining hypertrophic or hyperplastic adipose phenotypes. As the RNA-Seq was performed on whole SAT – comprising a heterogeneous mix of cell types – we next asked which adipose-resident cell populations preferentially express these morphology-associated genes. To address this, we clustered the 3727 candidate genes based on their relative expression across 16 human WAT cell types, using published single-cell transcriptomic data (*Figure 1*; *Figure 1—figure supplement 1*; *Emont et al., 2022*). As expected, the largest subset of morphology DEGs (n=730) was enriched in mature adipocytes, consistent with central roles for adipocytes in defining overall adipose morphology (*Figure 1—figure supplement 1*). Notably, we also identified 102 DEGs enriched in adipose stem and progenitor cells

(ASPCs) (*Figure 1*; *Figure 1—figure supplement 1*; *Supplementary file 2*). Given prior evidence that ASPC abundance and differentiation rate can influence adipose tissue expansion and morphology (*Gustafson et al., 2019*; *Hedbacker et al., 2020*), we hypothesised that these genes may function within ASPCs to regulate or establish adipose morphology.

## Analysis of candidate human adipose morphology genes using GWAS and cardiometabolic trait associations

Prior to functional analysis, we first explored genetic and cardiometabolic features of the 102 candidate genes. As an initial step, we examined overlap with body mass index (BMI)- and waist-to-hip ratio (WHR)-associated genome-wide association study (GWAS) genes, and found that 16 of the candidate morphology genes were also BMI GWAS genes (*TNRC6B, LRMDA, PRRX1, TSHZ2, PTPRD, PARD3, ADH1B, TBX15, MAGI2, RTL8A, CAST, BNC2, RERE, PTPRG, ADAMTSL1,* and *C1QBP*) and one gene, *HOXC4*, overlapped with WHR-associated loci; however, Fisher's exact test indicated these overlaps were not significantly enriched (*Figure 1—figure supplement 2*). We next clustered candidate genes based on their SAT expression correlations with 23 cardiometabolic traits from the METSIM study (*Civelek et al., 2017*). This revealed two distinct gene-trait signatures: one associated with a 'healthy' metabolic profile – characterised by higher hip circumference, fat-free mass, HDL-C, Muscle Insulin Sensitivity Index (MISI), and adiponectin levels, alongside lower BMI, WHR, total fatty acids, triacylglycerides, insulin, and HOMA-B – and another with an 'unhealthy' profile, showing the opposite trait pattern (*Figure 1—figure supplement 2*). Notably, genes associated with both hypertrophic and hyperplastic adipose morphology were found across both signatures (*Figure 1—figure supplement 2*), suggesting that these morphological phenotypes are not strictly aligned with traditional markers of metabolic health.

## Enrichment of cell differentiation and developmental genes among adipose morphology candidates

To explore functional themes within our gene set, we performed enrichment analysis to identify over-represented biological processes. This analysis revealed that 54 of the 102 ASPC-enriched genes were significantly associated with Gene Ontology (GO) terms related to cell differentiation (GO:0030154) and tissue development (GO:0048869, GO:0009888, GO:0032502) (*Figures 1 and 2a and b* and *Supplementary file 3*). Reasoning that they may play key roles in ASPC differentiation and/or the patterning of adipose tissue, we prioritised this subset of 54 genes for further analysis. To refine this list, we applied two complementary strategies. First, we ranked genes by their sequence conservation between humans and zebrafish (*Figure 2d*). Second, we assessed scRNA-Seq expression dynamics during progenitor differentiation to adipocytes (*Yang Loureiro et al., 2023*; *Figure 2d–f*). From this, we shortlisted 25 highly conserved genes that were predominantly expressed in either progenitors from undifferentiated samples, or within multipotent progenitors that persist following differentiation (*MGP*+ SWAT cells) (*Figure 2d–f* and *Table 1*). This final set included genes with established roles in adipocyte progenitors – such as *PRRX1* (*Liu et al., 2022*), *TBX15* (*Pan et al., 2021*), and *IRX1* (*Han et al., 2022*) – though their specific contributions to adipose morphology have not yet been characterised.

## An image-based pipeline to quantify LD size, number, and spatial patterning in zebrafish subcutaneous adipose

To investigate the function of the candidate morphology genes, we developed an image-based phenotyping pipeline in zebrafish. Building on previous studies where Nile Red fluorescence was used to quantify whole-animal adiposity and perform targeted CRISPR screens for regulators of total body fat (*Minchin and Rawls, 2017*; *Vishvanath and Gupta, 2019*), we refined this approach to capture not only global adiposity data but also high-resolution images of Nile Red-positive LDs. This enabled detailed quantification of LD size, number, and spatial patterning within zebrafish SAT (*Figure 3a–d*). High-magnification images were acquired with a pixel size between 1 and 2 μm$^2$, allowing individual LDs to be clearly resolved within zebrafish SAT (*Figure 3a–d*). Although very small LDs within multilocular adipocytes fall below our detection threshold, this resolution reliably captures the larger LDs characteristic of mature adipocytes (*Figure 3a–d*). Using this pipeline, we characterised SAT development in 107 wild-type zebrafish from three independent clutches (*Figure 1e–g*). Representative LD masks

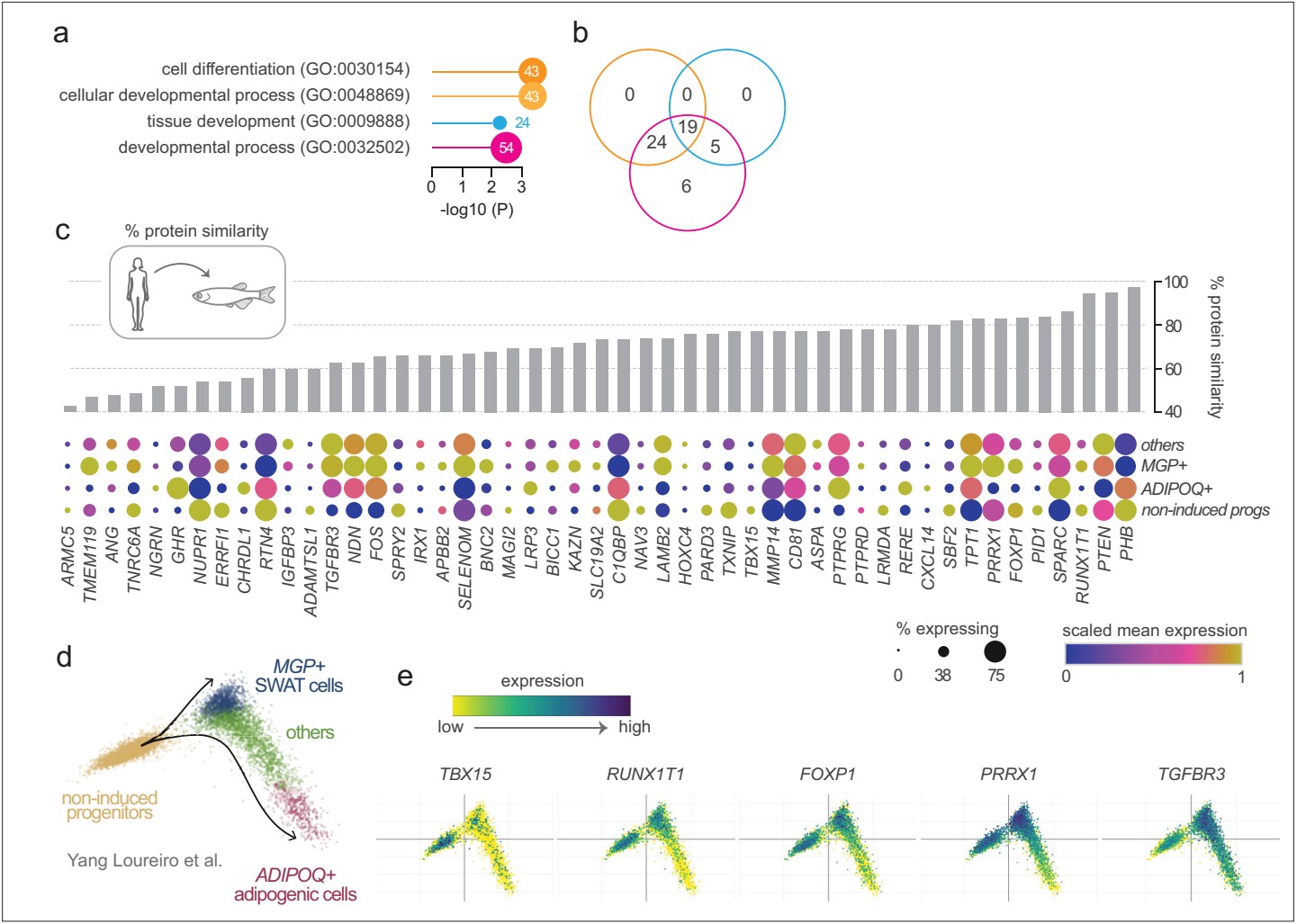

**Figure 2.** Prioritisation of candidate adipose morphology genes based on conservation to zebrafish and expression dynamics during progenitor differentiation. (**a**) Gene Ontology (GO) term enrichment analysis of the 102 candidate morphology genes enriched in adipose stem and progenitor cells. Four GO terms related to development and differentiation were significantly enriched, encompassing 54 genes. (**b**) Venn diagram showing gene overlap between the enriched GO terms. Cell differentiation (GO:0030154) and cellular developmental process (GO:0048869) had 100% overlap. (**c**) Protein conservation and expression during adipocyte progenitor differentiation for each candidate gene. Grey bars denote percentage protein similarity between human and zebrafish orthologues. Dot plots show scaled mean expression (colour) and percentage of cells expressing each gene (dot size) across four cell populations identified during in vitro progenitor differentiation: non-induced progenitors, *MGP+* SWAT cells, *ADIPOQ+* adipogenic cells and others (*Yang Loureiro et al., 2023*). Genes are ordered by protein similarity. (**d**) PCA projection of single cells coloured by cluster designation from *Yang Loureiro et al., 2023*. (**e**) Expression of selected candidate morphology genes projected onto the PCA embedding, illustrating enrichment during specific stages of adipocyte differentiation.

illustrating the morphological progression of SAT expansion are shown in *Figure 3e*. Consistent with previous observations, LDs are first deposited at the anterior end of the adipose depot and new LDs are added in both posterior and dorsoventral directions (*Figure 3e*). To spatially track this progression, we annotated the most anterior LD and defined successive 200 µm strata extending posteriorly from this reference point (*Figure 3e*). This stratification enabled comparisons of LD morphology and number at anatomically and developmentally equivalent positions across individuals. As zebrafish grew from 7 to 12 mm standard length (SL), we observed limited increases in LD number within the anterior-most stratum (stratum 1) (*Figure 3e and f*). Instead, new LDs accumulated in more posterior regions, particularly in strata 3 and 4, which exhibited the greatest LD number and overall growth (*Figure 3e and f*). Concurrently, LDs increased in size, reaching an average diameter of ~65 µm by 11–12 mm SL (*Figure 3e and g*). Notably, posterior strata – containing more recently formed LDs – appeared to be in the process of enlarging towards this consistent size benchmark (*Figure 3e and g*).

**Table 1.** Zebrafish gene targets and adipose morphology statistics.

Adjusted p-values after Benjamini-Hochberg false discovery rate (FDR) correction for 21 statistical tests. KS = Kolmogorov-Smirnov test. LMM = linear mixed model with experiment as random effect. Values in bold indicate statistical significance. Gene names in bold indicate robust phenotypes (significant in both stratified KS and LMM). Stratified KS tests were performed within each experiment and combined using Fisher's method. Genes with only one replicate (*aspa*, *lamb2*) or which showed lethality (*phb*, *rerea*) are not included in this table.

| Gene | Reps | n (ctrl/mut) | KS test | | | LMM | | Direction | Power (d=0.8) |
|---|---|---|---|---|---|---|---|---|---|
| | | | Effect (µm) | Pooled (FDR adj. p value) | Stratified (FDR adj. p value) | Effect (%) | FDR adj. p value | | |
| *txnipa* | 3 | 45/30 | +4.6 | 0.066 | **1.41e-04** | +19.9% | **1.63e-04** | Hypertrophic | 92% |
| *mmp14b* | 2 | 34/30 | +5.4 | **0.002** | **0.005** | +15.8% | **1.93e-04** | Hypertrophic | 88% |
| *foxp1b* | 2 | 42/45 | +8.8 | 0.064 | **0.019** | +17.0% | **0.049** | Hypertrophic | 96% |
| *ptprdb* | 2 | 35/24 | +4.0 | 0.080 | 0.057 | +18.5% | 0.087 | Hypertrophic | 84% |
| *cxcl14* | 2 | 27/24 | −4.4 | 0.522 | 0.077 | −8.7% | 0.348 | Hyperplastic | 80% |
| *prrx1b* | 3 | 65/60 | −3.5 | 0.178 | 0.077 | −5.9% | 0.358 | Hyperplastic | 99% |
| *ptenb* | 2 | 27/25 | −4.7 | 0.054 | 0.077 | −7.9% | 0.411 | Hyperplastic | 81% |
| *pid1* | 2 | 27/26 | −2.7 | 0.869 | 0.204 | −6.5% | 0.565 | Hyperplastic | 81% |
| *nav3* | 3 | 49/42 | −2.4 | 0.194 | 0.562 | −6.1% | 0.650 | Hyperplastic | 96% |
| *sdk1a* | 2 | 29/25 | +0.6 | 0.869 | 0.482 | +7.6% | 0.694 | Hypertrophic | 82% |
| *runx1t1* | 2 | 29/22 | +1.1 | 0.706 | 0.795 | +7.6% | 0.736 | Hypertrophic | 79% |
| *sparc* | 2 | 24/23 | −2.1 | 0.286 | 0.553 | −7.1% | 0.736 | Hyperplastic | 77% |
| *kazna* | 2 | 36/23 | −3.1 | 0.706 | 0.148 | −4.7% | 0.759 | Hyperplastic | 84% |
| *tmem115* | 2 | 27/29 | −3.1 | 0.488 | **0.019** | −3.8% | 0.759 | Hyperplastic | 84% |
| *cd81a* | 2 | 24/26 | +0.0 | 0.862 | 0.064 | −5.0% | 0.759 | Hypertrophic | 79% |
| *lrmda* | 2 | 27/22 | −0.4 | 0.527 | 0.077 | −2.7% | 0.778 | Hyperplastic | 78% |
| *tbx15* | 3 | 37/26 | +3.4 | 0.706 | **0.019** | +3.7% | 0.778 | Hypertrophic | 87% |
| *cacnb4a* | 2 | 29/27 | +0.9 | 0.706 | 0.852 | +2.6% | 0.804 | Hypertrophic | 84% |
| *pard3ab* | 2 | 36/32 | −1.5 | 0.869 | 0.750 | +2.1% | 0.804 | Hypertrophic | 90% |
| *ptprga* | 2 | 35/35 | −1.4 | 0.269 | 0.077 | −1.6% | 0.804 | Hyperplastic | 91% |
| *srpx* | 2 | 36/20 | −5.7 | 0.155 | 0.204 | −0.9% | 0.836 | Hyperplastic | 80% |

Together, these findings show that during these relatively young juvenile stages, LDs expand rapidly in size but ultimately stabilise at an average diameter of ~65 µm, reflecting a potential upper limit of LD growth at this developmental stage.

## Quantification of adipose morphology in juvenile zebrafish

As LD size, number, and spatial dynamics can be robustly captured in juvenile zebrafish, we next developed methodology to quantify adipose morphology – specifically hyperplastic versus hypertrophic patterns – in a format scalable for CRISPR-based screening. Our approach was conceptually based on previously described methods for assessing human adipose morphology (*Arner et al., 2010*). To establish a baseline, we quantified SAT LD number and size within 194 wild-type fish (including the 107 animals analysed in *Figure 3*). We observed a strong curvilinear relationship between mean LD diameter and the total depot size ($R^2$=0.85) (*Figure 4a*). This relationship also held true for other LD size metrics (including area and perimeter), further suggesting that as the zebrafish SAT expands, LDs initially grow rapidly in size but then plateau as they reach a steady-state size and hypertrophic growth slows relative to the overall size of the depot. Following the strategy of *Arner et al., 2010*, we calculated a 'morphology value' for each individual based on the deviation of each fish from the fitted

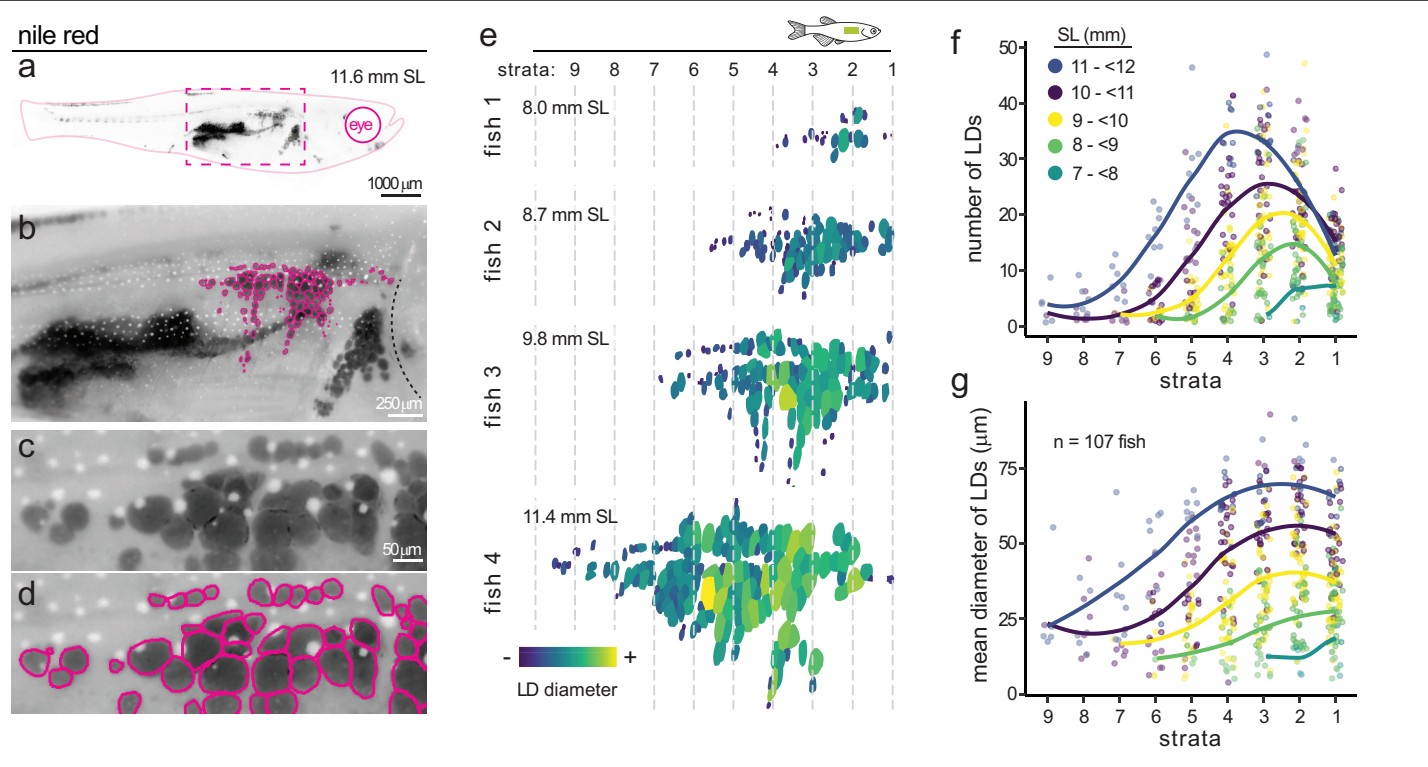

**Figure 3.** Spatial dynamics of subcutaneous adipose growth in zebrafish. (**a**) Nile Red-stained zebrafish at 11.6 mm standard length (SL). Black signal denotes Nile Red-positive neutral lipid within adipose tissue. Magenta dashed box indicates the region shown in (**b**). (**b**) Higher magnification of the subcutaneous adipose depot. Magenta outlines highlight segmented lipid droplets (LDs). Dashed line indicates the operculum. (**c**) Zoomed view of individual subcutaneous adipose LDs. White dots are melanosome pigment granules. (**d**) LD segmentation masks (magenta outlines) corresponding to (**c**). (**e**) Segmented subcutaneous adipose LDs from four representative zebrafish of increasing body size, colour-coded by LD diameter. Fish sizes are shown in mm SL. Dashed lines demarcate strata defined at 200 µm intervals from the most anterior LD. (**f**) Number of LDs per stratum as a function of zebrafish body size. Fish were grouped into five SL categories (colour-coded). Lines represent fitted curves; dots represent individual fish. (**g**) Mean LD diameter per stratum as a function of zebrafish body size (n=107 fish). Lines and colour coding as in (**f**).

curve (**Figure 4b**). Morphology values were normally distributed around zero, with approximately equal proportions of individuals exhibiting hyperplastic (negative deviation) or hypertrophic (positive deviation) SAT (**Figure 4b**). Notably, morphology values were inversely correlated with LD number (p=$3.9 \times 10^{-6}$), such that hypertrophic individuals had fewer, larger LDs while hyperplastic individuals had more numerous, smaller LDs (**Figure 4c**). Together, these data show that morphology values can be applied to zebrafish SAT and used to quantify hyperplastic/hypertrophic patterning.

## Morphology values can be used to assess differences in F0 zebrafish CRISPR mutants

We next tested whether this metric could detect known regulators of adipose morphology. As a positive control, we targeted *growth hormone 1* (*gh1*) using established F0 CRISPR mutant methodology. Consistent with previous findings that *gh1* mutants display hypertrophic SAT LDs (**McMenamin et al., 2013**), *gh1* F0 CRISPR mutants displayed significantly larger LDs and an upward shift in morphology values (p=0.04) (**Figure 4—figure supplement 1**). Notably, this effect was detected in only nine *gh1*-targeted fish, corresponding to a large effect size (Cohen's d=1.06) (**Figure 4—figure supplement 1**). Power analysis based on this dataset indicated that similar effects could be detected with 80% power using as few as 11 fish. Moreover, detecting medium-sized effects (Cohen's d=0.6–0.7) would require only 23–31 mutant fish. Together, these results demonstrate that image-based profiling of adipose morphology is a sensitive and scalable approach for in vivo genetic screens aimed at identifying regulators of hyperplastic and hypertrophic adipose growth.

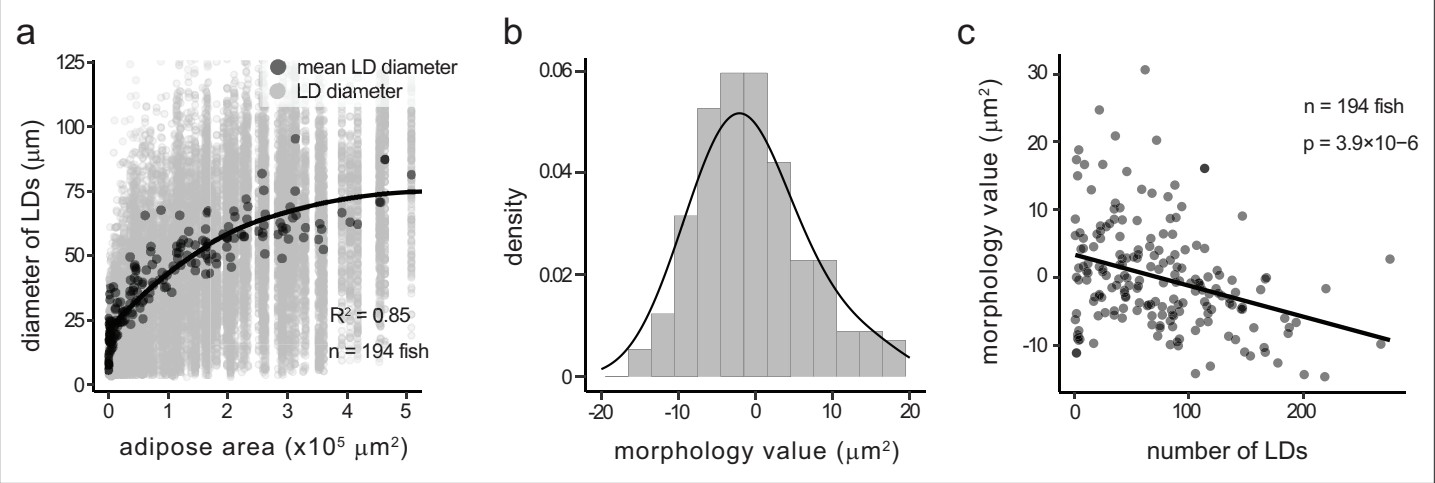

**Figure 4.** Derivation of the adipose morphology value from the relationship between lipid droplet size and adipose depot size. (**a**) Diameter of subcutaneous adipose lipid droplets (LDs) plotted against total depot area. Light grey dots represent individual LDs; dark grey dots represent the mean LD diameter per fish. Black line was fitted using a generalised additive model (GAM; $R^2$=0.85, n=194 fish). (**b**) Distribution of morphology values, calculated as the residual deviation of each fish's mean LD diameter from the GAM fit in (**a**). Positive values indicate larger LDs than expected (hypertrophic); negative values indicate smaller LDs than expected (hyperplastic). Black line shows a normal density fit. (**c**) Morphology value plotted against number of LDs per fish, showing an inverse relationship (linear model, p=3.9 × 10⁻⁶, n=194 fish). Fish with hypertrophic morphology tend to have fewer, larger LDs, while fish with hyperplastic morphology have more numerous, smaller LDs.

The online version of this article includes the following figure supplement(s) for figure 4:

**Figure supplement 1.** Image-based adipose morphology profiling detects hypertrophic lipid droplet (LD) expansion in *gh1* CRISPR mutants.

## A targeted CRISPR screen identified three genes that induce hypertrophic morphology in subcutaneous adipose

Following validation of our morphology quantification approach using *gh1*, we applied the same F0 CRISPR methodology and image-based pipeline to functionally assess all 25 conserved candidate genes identified from human adipose tissue datasets. In total, we screened 1371 juvenile zebrafish comprising 738 Cas9-only controls and 633 gene-targeted mutants across 2–3 independent experimental replicates per gene (*Figure 5a–c* and *Table 1*). Two targets, *phb* and *rerea*, were lethal prior to 5 days post fertilisation (dpf) and therefore excluded from morphological analysis. Of the remaining 23 genes, none showed significant effects on SL after controlling for batch effects, indicating that CRISPR targeting did not cause generalised growth defects (*Figure 5d*). One gene (*kazna*) showed significantly increased total adiposity (*Figure 5e*).

To quantify SAT morphology and classify hyperplastic or hypertrophic profiles, we applied two complementary statistical approaches. First, we calculated morphology values for each fish and compared mutant versus control morphology value distributions using the Kolmogorov-Smirnov (KS) test. This non-parametric test does not assume a specific distribution and is sensitive to shape-based distributional changes. To ensure effects were consistent across experimental batches rather than driven by between-experiment variability, we performed both pooled KS tests (combining data across all experiments) and stratified KS tests (testing within each experiment separately, then combining p-values using Fisher's method). Second, we applied linear mixed models (LMMs) to evaluate the relationship between LD size and total SAT area, whilst accounting for replicate-specific variability. The LMMs allowed us to test whether LDs scaled normally with tissue size (slope) or differed in size at any given adipose area (intercept). After correcting for multiple tests using the Benjamini-Hochberg procedure, we identified three genes with robust hypertrophic morphology phenotypes validated by both stratified KS tests (which control for batch effects by testing within each experiment) and LMMs: *txnipa* (+19.9% increase in LD diameter, stratified KS adj.p=1.41 × 10⁻⁴, LMM adj.p=1.63 × 10⁻⁴), *mmp14b* (+15.8%, stratified KS adj.p=0.005, LMM adj.p=1.93 × 10⁻⁴), and *foxp1b* (+17.0%, stratified KS adj.p=0.019, LMM adj.p=0.049) (*Figure 5*, *Table 1*). Two genes (*tmem115* and *tbx15*) showed significant distributional shifts in stratified KS tests (adj.p=0.019 for both) but non-significant LMM intercept effects, suggesting altered variance or size-dependent phenotypes rather than uniform

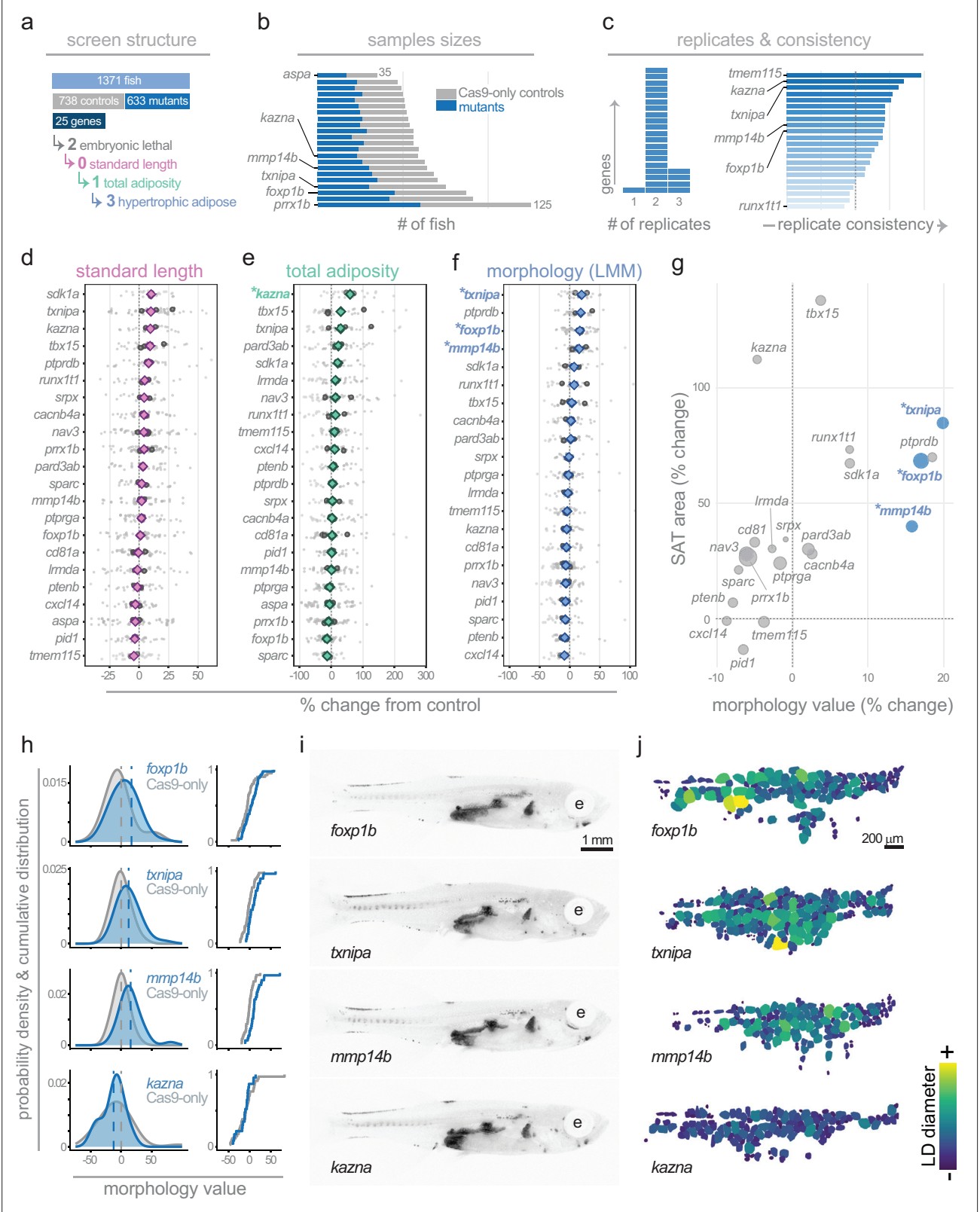

**Figure 5.** A targeted CRISPR screen in zebrafish identifies genes regulating hypertrophic or hyperplastic adipose morphology. (**a**) Overview of CRISPR screen structure. In total, 1371 fish were screened (738 Cas9-only controls and 633 mutants) across 25 candidate genes. Two genes (*phb* and *rerea*) were lethal before 5 days post fertilisation (dpf). Of the remaining genes, zero altered standard length, one (*kazna*) significantly increased total adiposity, and three (*txnipa*, *mmp14b* and *foxp1b*) produced significant hypertrophic subcutaneous adipose morphology. (**b**) Sample sizes per gene.

*Figure 5 continued on next page*

*Figure 5 continued*

Grey bars indicate Cas9-only controls; blue bars indicate mutants. *aspa* had the smallest and *prrx1b* the largest sample size. (**c**) Left panel: number of experimental replicates per gene. The majority of targets had two replicates: *aspa* had one replicate, and *nav3*, *prrx1b*, *tbx15*, and *txnipa* had three. Right panel: replicate consistency scores, representing agreement in effect direction and magnitude across replicates. Genes are ordered by consistency score. (**d–f**) Forest plots showing effect sizes (% change from Cas9-only controls) for standard length (d, magenta), total adiposity (e, green), and morphology value (f, blue). Small grey dots represent individual mutant fish; medium grey circles represent experiment means; coloured diamonds represent gene-level estimates from linear mixed models (LMMs) with experiment as a random effect. Morphology value represents the deviation in lipid droplet (LD) Feret diameter from that expected given total depot area, with positive values indicating hypertrophy and negative values indicating hyperplasia. Asterisks denote BH-adjusted p<0.05. (**g**) Scatter plot of depot (subcutaneous adipose tissue [SAT]) lipid area (% change from controls) versus morphology value (LMM estimate, % change from controls) for each gene. Point size reflects sample size. Quadrants indicate the combination of depot size and cell morphology phenotype (e.g. bigger depot with hypertrophic cells). (**h**) Probability density functions (left) and cumulative distribution functions (right) of morphology values for *foxp1b*, *txnipa*, *mmp14b*, and *kazna* F0 mutants (blue) compared with matched Cas9-only controls (grey). Dashed lines indicate group means. (**i**) Representative Nile Red images of *foxp1b*, *txnipa*, *mmp14b*, and *kazna* F0 mutants showing total adiposity. e, eye. Scale bar: 1 mm. (**j**) Segmented subcutaneous adipose LDs from representative mutant fish, colour-coded by LD diameter. Scale bar: 200 μm.

shifts in LD size (*Table 1*). Indeed, *tbx15* mutants displayed a significantly steeper size-scaling slope (+0.107, adj.p=0.026), indicating that any hypertrophic tendency is more pronounced in larger fish (*Table 1*). Three additional genes showed nominally significant effects in pooled KS tests prior to false discovery rate (FDR) correction but were not validated by stratified within-experiment analysis: *ptprdb* (hypertrophic, +18.5%), *ptenb* (hyperplastic, −7.9%), and *srpx* (hyperplastic, −0.9%) (*Table 1*). These represent suggestive hits that may warrant further investigation in stable mutant lines. Together, these findings demonstrate image-based profiling is an effective and scalable approach for identifying regulators of adipose morphology.

## A hypertrophic growth bias of subcutaneous adipose in *foxp1b* and double *foxp1a;foxp1b* zebrafish mutants

Notably, targeting *foxp1b* (one of two zebrafish *foxp1* genes) induced hypertrophic SAT morphology (*Figure 5*). Foxp1 is required for maintaining undifferentiated and self-renewing stem and progenitor cells in multiple tissue compartments (*Fu et al., 2018*; *Li et al., 2017*; *Naudin et al., 2017*; *Pearson et al., 2020*; *Zhao et al., 2015*). Further, in mouse, Foxp1 inhibits brown adipogenesis and browning of white adipocytes (*Liu et al., 2019*). However, roles for Foxp1 in establishing hyperplastic/hypertrophic morphology of adipose remain unknown. To further study the role of Foxp1, we generated new loss-of-function alleles for both *foxp1* genes (*foxp1a* and *foxp1b*) in zebrafish (*Figure 6a*). We targeted the highly conserved Forkhead domain (FHD) (*Figure 6b*), which is essential for DNA binding and therefore Foxp1 transcriptional activity (*Dai et al., 2021*; *Stroud et al., 2006*; *Figure 6b*). We generated two alleles – *foxp1a*[ed116] and *foxp1b*[ed125] – which each contained a frameshift and premature stop codon within the FHD (*Figure 6b*). qRT-PCR revealed reduced *foxp1a* or *foxp1b* mRNA in the respective single mutants, with no apparent mRNA upregulation of the paralogous *foxp1* gene, or other members of the Foxp family (*Figure 6—figure supplement 1*). Western blots revealed a significant reduction of Foxp1 protein in each of the single mutants, and in *foxp1a;foxp1b* double mutants, suggesting severe loss-of-function alleles (*Figure 6c*, *Figure 6—figure supplement 1*). At approximately 1 month of age (~10 mm SL), *foxp1b*[ed125] and double *foxp1a;foxp1b* mutants were significantly smaller than either wild-type control or *foxp1a*[ed116] fish (*Figure 6e*). Analysis of baseline lipid storage using Nile Red revealed that *foxp1a;foxp1b* double mutants had significantly lower normalised adipose area than controls, while *foxp1b* single mutants showed a non-significant trend (*Figure 6d and f*). Critically, despite their smaller body size, both *foxp1b*[ed125] and *foxp1a;foxp1b* mutants had significantly larger LDs characteristic of hypertrophic morphology (*Figure 6g*). This dissociation between body size (smaller) and LD size (larger) argues against a simple developmental delay: if these mutants were merely developmentally delayed, we would expect smaller fish to have smaller LDs, as observed during normal development (*Figure 3g*). Instead, *foxp1b* mutants seem to display fundamentally altered adipose growth properties. Analysis of blood metabolites revealed *foxp1a;foxp1b* double mutants were hyperglycaemic (*Figure 6—figure supplement 2*), and strikingly, in *foxp1a;foxp1b* double mutants there was increased lipid accumulation in the liver (*Figure 6d*, *Figure 6—figure supplement 2*). At adulthood, fish size and lipid storage within adipose in *foxp1a;foxp1b* double mutants had largely caught up with wild-type control fish (adiposity at ~80% of wild-type levels)

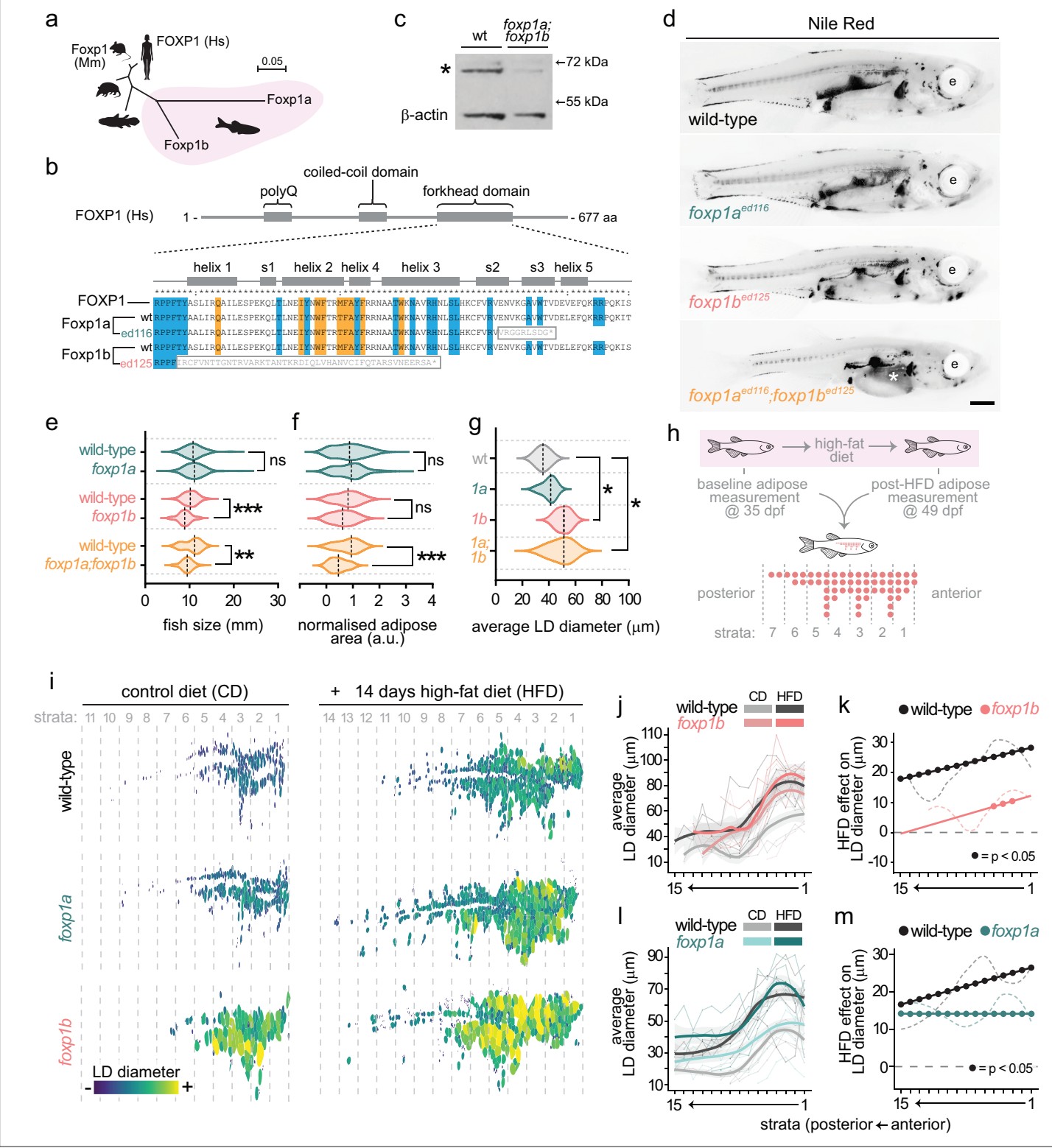

**Figure 6.** Stable *foxp1b* zebrafish mutants have hypertrophic adipose but undergo severely reduced hypertrophic remodelling in response to a high-fat diet (HFD). (**a**) Phylogenetic tree showing relatedness of zebrafish Foxp1a and Foxp1b amino acid sequences to human, mouse, opossum, and coelacanth Foxp1. Scale bar indicates substitutions per site. (**b**) Overview of human FOXP1 domain structure showing polyglutamine (polyQ), coiled-coil, and forkhead domains. Zoomed view of the DNA-binding forkhead domain showing structural features, including helices (helices 1–5) and beta-strands (s1–s3). Amino acids involved in DNA binding are highlighted in blue; residues at the FOXP domain-swapped dimer interface are highlighted in orange.

*Figure 6 continued on next page*

*Figure 6 continued*

Wild-type zebrafish Foxp1a and Foxp1b sequences are aligned to human FOXP1, along with the ed116 (foxp1a) and ed125 (foxp1b) mutant alleles. Grey boxes indicate the addition of nonsense peptide sequence followed by a premature stop codon. (**c**) Western blot showing reduction of Foxp1 protein in foxp1a;foxp1b double mutants compared to wild-type. β-Actin serves as a loading control. Asterisk indicates the Foxp1 band. (**d**) Nile Red fluorescence images showing adipose lipid distribution (black signal) in wild-type, *foxp1a^ed116^*, *foxp1b^ed125^*, and double *foxp1a^ed116^;foxp1b^ed125^* zebrafish mutants. e, eye. Asterisk indicates lipid accumulation within the liver in double mutants. Scale bar is 1 mm. (**e**) Violin plots of fish size (standard length, mm) in *foxp1a^ed116^*, *foxp1b^ed125^*, and double *foxp1a^ed116^;foxp1b^ed125^* mutants compared to wild-type siblings. (**f**) Violin plots of normalised adipose area in the same genotypes. (**g**) Violin plots of average lipid droplet (LD) diameter in the same genotypes. (**h**) Schematic of the HFD feeding experiment. Zebrafish were Nile Red-imaged at 35 days post fertilisation (dpf) to establish baseline adipose measurements, then subjected to a 14-day HFD (2 hr daily immersion in 5% chicken egg yolk) or control diet (2 hr daily immersion in system water), in addition to normal feeding. Post-diet Nile Red imaging was performed at 49 dpf. Subcutaneous adipose tissue was divided into anterior-posterior strata for spatial analysis. 200 μm strata are numbered anterior (1) to posterior. (**i**) Segmented subcutaneous adipose LDs from representative wild-type, *foxp1a*, and *foxp1b* fish on control diet (left) and after HFD (right), colour-coded by LD diameter. Strata boundaries (200 μm) are indicated by dashed lines. (**j**) Average LD diameter per stratum for wild-type and *foxp1b* fish on control diet (grey or salmon) and HFD (black or pink). Thin lines represent individual fish; thick lines represent group means. (**k**) HFD effect on LD diameter per stratum (HFD minus control diet) for wild-type (black) and *foxp1b* (pink). Filled circles indicate strata where the HFD effect is significant (BH-adjusted p<0.05). (**l**) As in (**J**), comparing wild-type and *foxp1a*. (**m**) As in (**k**), comparing wild-type (black) and *foxp1a* (teal). Statistical tests in (e–g) were one-way ANOVA followed by Tukey's HSD post hoc test. **p<0.01, ***p<0.001, ns = not significant.

The online version of this article includes the following source data and figure supplement(s) for figure 6:

**Source data 1.** Original western blot to verify Foxp1 knockdown in double *foxp1a^ed116^* and *foxp1b^ed125^* mutants.

**Source data 2.** Original western blot with bands highlighted to verify Foxp1 knockdown in double foxp1a^ed116^ and foxp1b^ed125^ mutants.

**Figure supplement 1.** Stable *foxp1a* and *foxp1b* mutant alleles lead to reduced transcript and protein expression.

**Figure supplement 1—source data 1.** Original western blot to verify Foxp1 knockdown in *foxp1a^ed116^* mutants.

**Figure supplement 1—source data 2.** Original western blot to verify Foxp1 knockdown in *foxp1b^ed125^* mutants.

**Figure supplement 2.** Metabolic phenotypes of *foxp1a*, *foxp1b*, and *foxp1a;foxp1b* mutant zebrafish following dietary challenge.

**Figure supplement 3.** Adiposity in adult *foxp1a^ed116^;foxp1b^ed125^* zebrafish mutants.

**Figure supplement 4.** Impaired adipose expansion in all *foxp1* mutant genotypes following high-fat diet (HFD).

**Figure supplement 5.** Spatial analysis of lipid droplet (LD) size along the anterior-posterior axis in foxp1a and foxp1b mutants following high-fat diet (HFD) challenge.

**Figure supplement 6.** Decomposition of diet and genotype effects on lipid droplet (LD) size across the anterior-posterior axis in foxp1a and foxp1b mutants.

---

(*Figure 6—figure supplement 3*). These findings show that stable *foxp1b* mutants have hypertrophic adipose along with metabolic dysfunction.

## Spatial analysis reveals profoundly blunted hypertrophic responses to HFD in *foxp1b* mutants

Our previous analyses showed that *foxp1b* mutants display a bias toward hypertrophic growth during developmental expansion of SAT. To determine whether this bias impacts the adipose response to dietary challenge, we subjected *foxp1a* and *foxp1b* mutants to a 2-week HFD, a condition previously shown to induce SAT expansion (*Minchin and Rawls, 2017*). All *foxp1* mutant genotypes exhibited reduced overall HFD-induced adipose expansion compared to control fish, indicating a general impairment in expansion capacity (*Figure 6—figure supplement 4*). To characterise the HFD response, we performed spatial analysis by binning the anterior-posterior axis into 200 μm strata, enabling comparison of LD morphology at anatomically equivalent positions (*Figure 6h–m*). Using paired analysis (same fish imaged on control diet at 35 days post fertilisation [dpf] and after 14 days HFD at 49 dpf) with linear mixed-effects models, we quantified diet effects across all strata. First, we confirmed that on a control diet, *foxp1b* mutants had significantly larger LDs than wild-type across strata 1–5 (ranging from +22.2 μm in stratum 1 to +17.8 μm in stratum 5; all FDR-adjusted p<0.05; *Figure 6i*). By contrast, *foxp1a* mutants showed no baseline hypertrophy at any stratum (all p>0.10; *Figure 6—figure supplements 5 and 6*). Second, spatial analysis revealed that *foxp1b* mutants show profoundly blunted hypertrophic responses to HFD. In wild-type fish, HFD induced robust LD hypertrophy across all 15 strata (range: +17.7 to +28.1 μm; all FDR-adjusted p<0.025; *Figure 6j*). However, *foxp1b* mutants showed a strikingly different pattern: in anterior/oldest strata (1–5), mutants showed significant but attenuated HFD responses (+7.7 to +12.2 μm; ~57% weaker than wild-type; FDR-adjusted p<0.05

for strata 3–5), while in posterior/newer strata (7–15), there was essentially no hypertrophic response (effect sizes declining from +6.8 µm to +0.4 µm; all FDR-adjusted p>0.068; *Figure 6j*). Overall, averaging across all strata, wild-type LDs increased by +24.4 µm in response to HFD (p<0.0001), whereas *foxp1b* mutant LDs increased by only +7.7 µm (p=0.020) – representing a 68% reduction in hypertrophic capacity (*Figure 6k*). *foxp1a* mutants had no baseline hypertrophy on control diet (all strata p>0.10; *Figure 6l*), but still exhibited significantly blunted HFD responses (~35% reduction compared to wild-type; p=0.023; *Figure 6m*), demonstrating that an impaired adaptive response can occur independently of baseline LD size. Interestingly, *foxp1a* mutants also lost the spatial gradient of HFD responsiveness observed in wild-type fish (uniform +14.4 µm across all strata versus wild-type range of +16.6 to +26.4 µm anteriorly to posteriorly), suggesting Foxp1a may regulate spatial coordination within the tissue. Together, these spatial analyses reveal that *foxp1b* mutants have intrinsically impaired capacity for diet-induced hypertrophic remodelling, while *foxp1a* mutants show a distinct phenotype affecting the magnitude and spatial patterning of adaptive responses. These findings demonstrate that Foxp1 paralogues play non-redundant roles in regulating both constitutive adipose morphology and adaptive responses to nutritional challenge.

## Discussion

In this study, we (i) identified candidate human genes implicated in hypertrophic and hyperplastic adipose morphology, (ii) established an image-based profiling method for assessing adipose morphology in zebrafish, (iii) conducted an in vivo CRISPR screen to functionally evaluate the roles of 25 candidate genes with rigorous statistical validation, (iv) identified three genes that induce hypertrophic adipose morphology, (v) generated stable *foxp1a* and *foxp1b* zebrafish mutant lines that recapitulate phenotypes observed in the F0 CRISPR experiments and revealed paralogue-specific roles in adipose biology, and (vi) discovered through spatial analysis that *foxp1b* mutants have intrinsically impaired capacity for diet-induced hypertrophic remodelling. Together, this work demonstrates a scalable, rigorous screening platform to identify regulators of adipose morphology with clear translational potential.

### Candidate gene identification from human data

We employed a multi-step process to identify candidate morphology genes from human transcriptomic data. As GWAS for adipose morphology has only been conducted in relatively small cohorts (*Glastonbury et al., 2020*; *Kulyté et al., 2022*), we instead used published data identifying DEGs enriched in SAT characterised by large or small adipocytes (*Honecker et al., 2022*). Clustering of the morphology DEG expression correlations with cardiometabolic traits separated genes into 'healthy' and 'unhealthy' profiles (*Figure 1—figure supplement 2*); genes associated with larger adipocytes generally aligned with adverse metabolic traits such as increased BMI and WHR in accordance with adipocyte size increasing with BMI (*Ye et al., 2022*). However, this pattern was not absolute – some candidates diverged from this axis (*Figure 1—figure supplement 2*), representing potentially interesting regulators of adipose morphology that are uncoupled from traditional metabolic or disease states. We focused on candidate genes highly expressed in ASPCs, reasoning that progenitor function influences both hyperplastic potential and overall remodelling capacity. However, we note that adipose remodelling is likely determined by multiple cell types beyond progenitors, including the adipose microenvironment (*Jeffery et al., 2016*; *Minchin et al., 2015*).

### An image-based platform for profiling adipose morphology

We established a rapid, scalable, stereomicroscope-based imaging pipeline for profiling zebrafish adipose morphology. Our imaging resolution was sufficient to detect large LDs but not smaller secondary LDs found in zebrafish adipocytes (*Minchin et al., 2015*), a phenomenon also observed in mouse adipocytes (*Chau et al., 2014*). Nevertheless, our methodology reliably captured primary LDs and allowed us to use LD size as a proxy for overall adipocyte size. We adapted the morphology value framework developed by the Arner lab for human adipose (*Li et al., 2017*) and observed strikingly similar relationships in zebrafish SAT, including a curvilinear relationship between LD size and depot size and an inverse correlation between LD number and LD size. These parallels suggest that

the fundamental growth dynamics of zebrafish adipose are conserved with mammals, supporting zebrafish as a relevant model for studying adipose biology.

## Statistical validation and batch effect control

A key advance of our platform is the implementation of rigorous statistical validation to control for batch effects – a common source of false positives in genetic screens. We employed two complementary approaches: KS tests to compare morphology value distributions and LMMs to assess LD size while accounting for experimental batch as a random effect. Critically, we performed both pooled KS tests (combining data across experiments) and stratified KS tests (testing within each experiment separately, then combining p-values using Fisher's method). This stratified approach validates that phenotypic effects are reproducible within independent experiments rather than driven by between-batch variability. Genes were considered robust hits only if significant in both stratified KS and LMM analyses after FDR correction – a conservative threshold that substantially reduces false discovery risk. Using these criteria, we identified three genes (*txnipa*, *mmp14b*, *foxp1b*) as robust hypertrophic hits. Three additional genes (*ptprdb*, *ptenb*, *srpx*) showed suggestive effects in pooled analyses but were not validated by stratified tests, indicating potential batch artifacts or weaker effects requiring larger sample sizes. Two genes (*tbx15*, *tmem115*) showed significant distributional shifts in stratified KS without corresponding LMM intercept effects; notably, *tbx15* displayed a significant slope effect, indicating size-dependent rather than uniform phenotypes. This statistical framework – combining distribution-sensitive tests with mixed models and stratified validation – provides a template for future high-throughput screens where batch effects are unavoidable.

## Functional insights from screen hits

*Txnipa (txnipa)* F0 CRISPR mutants displayed hypertrophic morphology. TXNIP (thioredoxin-interacting protein) is a well-characterised regulator of oxidative stress, metabolism, and inflammation that binds and inhibits thioredoxin, leading to increased production of reactive oxygen species and oxidative stress (*Choi and Park, 2023*). TXNIP links nutrient sensing to inflammatory signalling by responding to glucose and regulating NLRP3 inflammasome activation (*Choi and Park, 2023*). The ROS/TXNIP/NLRP3 pathway impairs insulin signalling, a major driver of adipogenesis and hyperplastic adipose growth (*Russell-Guzmán et al., 2024*), potentially explaining why *txnipa* loss biases toward hypertrophy. Additionally, vitamin D increases TXNIP expression (*Hamilton et al., 2014*) and plays a major role in energy storage in zebrafish (*Freeburg et al., 2024*), suggesting nutrient-sensing pathways may coordinate morphological outcomes.

*Mmp14b* F0 zebrafish mutants also exhibited hypertrophic adipose. MMP14 (matrix metallopeptidase 14) is a pericellular collagenase and plays a key role in ECM remodelling. In mice, MMP14 deficiency stiffens the adipose matrix and restricts expansion, resulting in smaller adipocytes (*Chun et al., 2006*), while overexpression of Mmp14 increased the size of adipocytes (*Li et al., 2020*). Our zebrafish findings – hypertrophy upon mmp14b loss – appear to contrast with these mouse phenotypes and warrant further investigation. Notably, zebrafish possess two *mmp14* paralogues; therefore, functional redundancy or divergence between paralogues may underlie species-specific differences. Future studies targeting both paralogues simultaneously would clarify this.

## Paralogue selection: strengths and limitations

Our screen targeted single paralogues based on DIOPT orthology scores, selecting the zebrafish gene with highest predicted orthology to each human candidate. This approach has important limitations: orthology scores predict sequence conservation but not necessarily functional equivalence, meaning the 'most orthologous' paralogue may not carry the relevant adipose function. Indeed, our three candidate hit genes have multiple zebrafish paralogues (FOXP1, TXNIP, MMP14), and testing only one paralogue means we may have missed genuine hits where the alternative paralogue is functionally relevant. Our Foxp1 findings both validate this approach and reveal its limitations. The higher-scoring paralogue *foxp1b* (DIOPT score 13/19) showed the more severe baseline phenotype, consistent with our prioritisation. However, *foxp1a* (DIOPT score 5/19), tested subsequently, revealed a distinct and biologically significant phenotype affecting spatial patterning of HFD responses – a finding that would have been missed had we not pursued secondary validation. For future screens where comprehensive

hit identification is the goal, multiplexed F0 targeting of all paralogues may be valuable (67), though this approach may complicate interpretation of paralogue-specific phenotypes.

## Non-redundant roles for Foxp1 paralogues in adipose biology

*Foxp1* is known to maintain stem and progenitor populations across multiple tissues, including the hair follicle, haematopoietic niche, mammary gland, and cerebral cortex (*Fu et al., 2018*; *Naudin et al., 2017*; *Pearson et al., 2020*; *Zhao et al., 2015*). We hypothesised that Foxp1 may be required in adipose progenitors for hyperplastic growth, with its loss resulting in compensatory hypertrophy. *Foxp1b* mutants displayed constitutive hypertrophy at baseline – smaller fish with paradoxically larger LDs – along with a reduced hypertrophic response to HFD together with reduced adipose coverage, hyperglycaemia, and hepatic steatosis in double mutants. In contrast, *foxp1a* mutants showed no baseline morphological defects but also exhibited impaired adaptive responses to HFD and a break-down in anterior-posterior patterning in the adipose tissue. These data suggest both *foxp1* paralogues regulate adaptive responses to HFD, but also regulate distinct aspects of adipose biology: Foxp1b controlling constitutive morphology and Foxp1a regulating spatial patterning.

## Impaired adaptive capacity rather than developmental ceiling

A key finding from our spatial analysis is that *foxp1b* mutants have intrinsically impaired capacity for diet-induced hypertrophy, rather than having reached a developmental size limit. Several observations support this conclusion. First, the blunted HFD response in *foxp1b* mutants extended throughout the entire anterior-posterior axis (68% overall reduction), including posterior regions containing smaller, newer LDs. If anterior adipocytes had simply reached maximum size, we would expect normal responses posteriorly – but instead we observed complete loss of response in posterior strata. Second, and most strikingly, wild-type LDs on HFD actually surpassed *foxp1b* mutant sizes in posterior strata, demonstrating that these sizes are not developmentally limiting. Third, *foxp1a* mutants showed blunted HFD responses (~35% reduction) despite having no baseline hypertrophy, proving that impaired adaptive capacity can occur independently of baseline LD size. These findings have important implications. They suggest that both *foxp1a* and *foxp1b* mutants may fail to perceive or respond to HFD-induced molecular cues, potentially implicating Foxp1 in metabolic sensing pathways (66). The distinct *foxp1a* phenotype – loss of spatial gradients in HFD responsiveness – suggests this paralogue may coordinate regional heterogeneity within adipose tissue. Testing whether other hypertrophic mutants identified in our screen (*txnipa*, *mmp14b*) also show impaired adaptive responses would determine whether this reflects a general principle linking developmental morphology to later plasticity.

## Comparison with mammalian Foxp1 studies

In mouse, adipose-specific Foxp1 deletion increases browning of WAT and enhances thermogenesis, protecting against diet-induced obesity (*Liu et al., 2019*). Our zebrafish findings – where whole-organism *foxp1b* loss causes metabolic dysfunction rather than protection – appear contradictory but likely reflect important experimental differences. First, Liu et al. studied browning-prone inguinal WAT, whereas zebrafish adipose, as in an ectotherm, is not traditionally considered thermogenic – though recent evidence shows zebrafish epicardial adipose can be thermogenic (*Morocho-Jaramillo et al., 2024*). In browning-resistant depots, more commonly observed in humans, Foxp1 loss might produce phenotypes more similar to our zebrafish findings. Second, our whole-organism mutants also lack hepatic Foxp1 which regulates gluconeogenesis (*Zou et al., 2015*); the metabolic dysfunction we observe may reflect hepatic rather than adipose-autonomous effects. Tissue-specific approaches in zebrafish would help distinguish these contributions.

## Platform scalability and translational potential

Our screening platform offers several advantages for systematic functional genomics. First, it is highly scalable: we screened 1371 fish across 25 genes with sufficient statistical power (>80% for large effects) in each case. The imaging pipeline requires only standard stereomicroscopy, and analysis is fully auto-mated. Second, it is modular: the same fish can be assessed for multiple phenotypes (body size, total adiposity, morphology, spatial patterning), and the platform readily accommodates dietary or phar-macological challenges. Third, results translate meaningfully to mammalian biology, as demonstrated by the conservation of morphology value relationships. Compared to mammalian screening platforms,

zebrafish offer dramatically reduced costs and timelines while enabling whole-organism assessment of gene function. However, limitations include the lack of tissue-specific genetic tools comparable to Cre-lox systems, potential paralogue complications, and the possibility that some mammalian adipose biology (particularly thermogenesis) may not be fully conserved. Nevertheless, our identification of genes with established human disease relevance (*TXNIP*, *MMP14*, *FOXP1*) demonstrates clear translational line-of-sight from zebrafish discoveries to human adipose biology.

# Materials and methods

**Key resources table**

| Reagent type (species) or resource | Designation | Source or reference | Identifiers | Additional information |
|---|---|---|---|---|
| Strain, strain background (*Danio rerio*, juvenile fish prior to overt sex differentiation) | WIK | *Rauch et al., 1997* | ZFIN: ZDB-GENO-010531-2 | |
| Genetic reagent (*Danio rerio*) | *foxp1a* | This manuscript | ed116 | New zebrafish *foxp1a* mutant – 13 bp indel in forkhead domain |
| Genetic reagent (*Danio rerio*) | *foxp1b* | This manuscript | ed125 | New zebrafish *foxp1b* mutant – 4 bp indel in forkhead domain |
| Sequence-based reagent | *foxp1a* | IDT | CRISPR gRNA | GACGAGTGGAGAATGTGAAG |
| Sequence-based reagent | *foxp1b* | IDT | CRISPR gRNA | GATAACGAAGCATACGTGAA |
| Commercial assay or kit | KASP on demand genotyping assays | LGC Biosearch Technologies | *foxp1a*$^{ed116}$ and *foxp1b*$^{ed125}$ specific assays | |
| Commercial assay or kit | Glucose assay | Biovision | K606-100 | |
| Commercial assay or kit | Triacylglyceride assay | Biovision | K622-100 | |
| Commercial assay or kit | Cholesterol assay | Biovision | K623-100 | |
| Commercial assay or kit | Luna Universal qPCR Master Mix | NEB | Cat #: M3003L | |
| Sequence-based reagent | *foxp1a* | IDT | PCR primers for T7E1 assay (ed116) | Forward primer – GCCAGATTGGACTGGATGTT |
| Sequence-based reagent | *foxp1a* | IDT | PCR primers for T7E1 assay (ed116) | Reverse primer – TTATTTCCAGGCCATTCTGG |
| Sequence-based reagent | *foxp1b* | IDT | PCR primers for T7E1 assay (ed125) | Forward primer – TTCAGTTTCAGCTCCTTCCTTC |
| Sequence-based reagent | *foxp1b* | IDT | PCR primers for T7E1 assay (ed125) | Reverse primer – TGGAAGTCAAGCTACCAGCA |
| Antibody | Anti-FOXP1 polyclonal antibody (rabbit) | Thermo Fisher | Cat #: PA5-26848 | 1:1000 dilution |
| Antibody | Anti-β-Actin (ACTB) antibody (mouse) | Sigma-Aldrich | Cat #: A2228 | 1:25,000 dilution |
| Chemical compound, drug | Nile Red | Sigma-Aldrich | Cat #: 19123 | |
| Peptide, recombinant protein | Cas9 | New England Biolabs | Cat #: M0646T | |
| Peptide, recombinant protein | T7 Endonuclease I | New England Biolabs | Cat #: M0302S | |
| Software, algorithm | GProfiler | *Kolberg et al., 2023* | RRID:SCR_006809 | |
| Software, algorithm | Single Cell Portal | *Tarhan et al., 2023* | RRID:SCR_014816 | |
| Software, algorithm | Morpheus | *Gould, 2022* | RRID:SCR_014975 | |
| Software, algorithm | Heatmapper | *Babicki et al., 2016* | RRID:SCR_016974 | |
| Software, algorithm | GWAS Catalog | *Cerezo et al., 2025* | | https://www.ebi.ac.uk/gwas/ |

*Continued on next page*

*Continued*

| Reagent type (species) or resource | Designation | Source or reference | Identifiers | Additional information |
|---|---|---|---|---|
| Software, algorithm | DIOPT | *Hu et al., 2011* | | https://www.flyrnai.org/diopt |
| Software, algorithm | UCSC Genome Browser | *Casper et al., 2026* | RRID:SCR_005780 | |
| Software, algorithm | CHOPCHOP | *Labun et al., 2019* | RRID:SCR_015723 | |
| Software, algorithm | Cellpose | *Stringer et al., 2021* | RRID:SCR_022332 | |
| Software, algorithm | Fiji/ImageJ | *Schindelin et al., 2012* | RRID:SCR_002285 | |
| Software, algorithm | Napari | *Sofroniew et al., 2026* | RRID:SCR_022765 | |
| Software, algorithm | R Statistical software v. 4.5.1 | R Project for Statistical Computing | RRID:SCR_001905 | |

## Identification of DEGs associated with small or large adipocytes in human subcutaneous adipose

DEGs associated with large or small adipocytes in human SAT were obtained from *Honecker et al., 2022*. Briefly, 3727 genes were significantly associated with adipocyte area by both categorical and continuous statistical models (FDR<0.05) (*Honecker et al., 2022*). 2190 DEGs were associated with small adipocyte area, 1537 DEGs were associated with large adipocyte area (*Supplementary file 1*). Expression within human WAT cell types for each of the candidate morphology genes was determined using data from *Emont et al., 2022*, via the Single Cell Portal (https://singlecell.broadinstitute.org/single_cell). Scaled mean expression for each of the morphology DEGs was assessed within each of the 16 previously defined WAT cell clusters (ASPCs, mesothelium, pericyte, adipocyte, macrophage, endothelial, LEC, monocyte, T cells, dendritic cells, smooth muscle cells, B cells, mast cells, natural killer cells, endometrium, and neutrophils). Of the 3727 morphology genes, the following were removed: 410 genes without annotations in bulk RNA-Seq dataset, 250 genes not found in the Single Cell Portal dataset and 87 genes not expressed in the 16 WAT cell types. Hierarchical clustering of the remaining 2980 morphology genes based on WAT cell-type expression was performed in Morpheus (https://software.broadinstitute.org/morpheus) using an average linkage method and one minus Pearson correlation metric (*Figure 1—figure supplement 1*). Morphology genes enriched in ASPCs were identified as either (i) belonging to the 75 morphology DEGs within the ASPC cluster or (ii) one of the top 50 morphology DEGs with highest scaled mean expression in ASPCs (*Supplementary file 2*). Three genes were present in both categories (*CXCL14*, *KAZN*, and *SPARC*), resulting in 122 ASPC-enriched morphology DEGs (*Supplementary file 2*). Ribosomal genes were excluded along with genes that did not code for proteins, resulting in 102 candidate morphology genes (*Supplementary file 2*).

## Prioritisation and ranking of candidate adipose morphology genes

Enrichment analysis on the 102 candidate genes described above was performed using GProfiler (https://biit.cs.ut.ee/gprofiler/gost), with the 3727 original genes as a background gene set. Six enriched GO biological process gene sets were identified, comprising: three 'developmental' terms (GO:0048869, GO:0009888 and GO:0032502), one 'differentiation' term (GO:0030154), and two more generic 'process' terms (GO:0048869 and GO:0048523) (*Supplementary file 3*). For each of the 102 morphology genes, β coefficients between SAT expression levels and the 23 cardiometabolic traits from the METSIM study were clustered in Heatmapper (http://www.heatmapper.ca) using an average linkage clustering method and Pearson distance measurement method (*Supplementary file 2*). Correlations between SAT expression and 23 cardiometabolic traits from 770 males as part of the METSIM study were obtained from *Civelek et al., 2017*. The following genes were not found in the METSIM data: *EIF4EBP3*, *PDF*, *SELENOM,* and *SELENOS*. Candidate BMI and WHR GWAS genes were identified by (i) downloading summary statistics for BMI and WHR from the GWAS Catalog (https://www.ebi.ac.uk/gwas/), (ii) identifying unique variants associated with the traits at genome-wide significance ($p < 5 \times 10^{-8}$), (iii) calculating proxy variants in high linkage disequilibrium in CEU

population ($R^2$>0.8), (iv) calculating 'LD blocks' that encompass each lead and associated proxy variants, and (v) identifying genes that overlap with the LD blocks. The DIOPT Ortholog Finder (https://www.flyrnai.org/cgi-bin/DRSC_orthologs.pl) was used to identify zebrafish orthologs of the candidate human morphology genes based on the weighted score. The 'alignment & scores' function in DIOPT was used to calculate protein similarity between human and zebrafish genes. scRNA-Seq scaled mean expression and % of cells expressing candidate genes from *Yang Loureiro et al., 2023*, were obtained from the Single Cell Portal.

## Zebrafish husbandry and maintenance

Zebrafish were housed in the Queen's Medical Research Institute (QMRI) facility at the University of Edinburgh, UK. Standard husbandry was followed on a recirculating system and a 14 hr:10 hr light to dark cycle. Feeding regimen was as described in *Tandon et al., 2025*. The wild-type line used was WIKs maintained at the QMRI facility. Zebrafish experiments were conducted in accordance with the UK Animals (Scientific Procedures) Act 1986 under the project licence PP9112175.

## F0 CRISPR screen in zebrafish

We followed general CRISPR screen methodology (*Kroll et al., 2021*; *Varshney et al., 2015*; *Wu et al., 2018*). Between three and five guide RNAs (gRNAs) were designed to target each gene using the UCSC Genome browser track ZebrafishGenomics (*Varshney et al., 2015*) and CHOPCHOP software. gRNAs were selected based on off-target and efficiency scores, along with location in gene (targeting early exons, functional domains, and avoiding exons with lengths divisible by three). Pooled in vitro transcription of gRNAs was performed as per *Wu et al., 2018*, and RNA was cleaned using Zymoclean columns (Zymo Research, Cat. # R1013). Prior to injection, gRNAs were incubated with Cas9 protein (New England Biolabs, Cat. # M0646T) and heated for 5 min at 37°C. Mutagenesis of individual gRNAs was verified using T7E1 assays (New England Biolabs, Cat. # M0302S). gRNAs were injected into zebrafish embryos at the one-cell stage as per *Wu et al., 2018*, along with a Fast Green dye to screen for successfully injected embryos. Identical injection mixes lacking the gRNAs (Cas9-only control) were used for control groups. 50 injected embryos were placed per Petri dish and screened for survival at 5 dpf. At 5 dpf, injected larvae were placed in a small volume of system water in a 1 L nursery tank without running water. From 10 dpf, a slow flow of system water was introduced. Larvae were raised at a density of 25 per 1 L nursery tank, before being transferred to a 3 L tank at 21 dpf. Survival was further assessed at 21 dpf. Feeding regimen was as described for standard husbandry. At 36–42 dpf, fish were euthanised and Nile Red staining performed as described previously (*Minchin and Rawls, 2017*).

## Generation of *foxp1a* and *foxp1b* mutant zebrafish lines

*foxp1a* and *foxp1b* zebrafish mutant lines were made using standard CRISPR-Cas9 methods (*Tandon et al., 2025*). For both genes, we targeted the highly conserved DNA-binding FHD. The location and sequence of the CRISPR gRNAs were: *foxp1a* – 5'-GACGAGTGGAGAATGTGAAG-3' targeting the FHD in exon 14 and *foxp1b* – 5'-GATAACGAAGCATACGTGAA-3' targeting the FHD in exon 15. We generated two alleles – ed116 is a 13 bp indel in the FHD of *foxp1a* resulting in a frameshift and premature stop codon, and ed125 is a 4 bp indel in the FHD of *foxp1b* resulting in a frameshift and premature stop. Primers to amplify locus surrounding lesions for T7E1 assays or sequencing were: ed116 forward – 5'-GCCAGATTGGACTGGATGTT-3', ed116 reverse – 5'-TTATTTCCAGGCCATT CTGG-3', ed125 forward – 5'-TTCAGTTTCAGCTCCTTCCTTC-3', ed125 reverse – 5'-TGGAAGTC AAGCTACCAGCA-3'. To genotype ed116, primers were designed to recognise either the 13 bp indel or the wild-type sequence. Sequences were: forward – 5'-GCCAGATTGGACTGGATGTT-3', reverse - 5'-TTATTTCCAGGCCATTCTGG-3', wild-type – 5'-ACGAGTGGAGAATGTGAAGG-3', ed116 – 5'-AAACGGCCCCCTCGTACT-3'. For ed125, the following primers were used to amplify the locus: forward – 5'-TTCAGTTTCAGCTCCTTCCTTC-3', and reverse – 5'-TGGAAGTCAAGCTACCAGCA-3'. The PCR product was then digested with the restriction enzyme HpyCH4IV (New England Biolabs, Cat. # R0619S). Additionally, KASP genotyping assays were designed for the ed116 and ed125 alleles (KASP on demand, LGC Biosearch Technologies). Western blots were performed to assess the effect of mutations on Foxp1 protein. Protein was isolated from the caudal fin – a pool of seven fin samples was placed on dry ice in RIPA buffer (Thermo, Cat. # 8990) containing protease inhibitor (Thermo,

Cat. # A32953). Samples were homogenised with a TissueRupter, centrifuged for 15 min at 4°C, lysate removed and added to LDS blue (Novex, Cat. # B0008) with reducing agent (Novex, B0009), heated to 95°C for 10 min and then centrifuged for 1 min. Protein samples were run on 8% Bis-Tris gels (Invitrogen, NW00082BOX) and run together with protein ladder (Thermo Scientific, Cat. # 26619). Foxp1 antibody was PA5-26848 (Thermo Fisher) and β-actin antibody was Sigma, A2228. Blood assays were from Biovision (Glucose – K606-100, triacylglyceride – K622-100, cholesterol – K623-100) and performed as per the manufacturer's instructions with 1 μL of blood from an adult zebrafish used for the glucose and cholesterol assays, and 0.2 μL for the triacylglyceride assay. qRT-PCR was performed on a Roche LightCycler 480 using Luna reagents (New England Biolabs, Cat. # M3003L). cDNA was synthesised using Superscript IV (Invitrogen, Cat. # 18090010). Primers were: *foxp1a* forward 5'-GGCC ACTTTGAGGATGACTC-3', *foxp1a* reverse 5'-CCTCGCCACCTAAAACTCAG-3', *foxp1b* forward 5'-CATTGGCTCCTCTTTTACGC-3', *foxp1b* reverse 5'-ACAGCAACGGTAGTGACAGC-3', *bactin* forward 5'-GCCTCCGATCCAGACAGAGT-3', and *bactin* reverse 5'-TGACAGGATGCAGAAGGAGA-3'.

## HFD in zebrafish

A high-fat immersion diet was conducted on juvenile fish at 35 dpf. Genotyped fish were stained with Nile Red, imaged on a Leica M205 FCA stereoscope and recovered individually in a 12-well plate whilst the SL of each fish was measured. Based on SL, fish were placed into groups with equal average SL. Within groups, fish were randomly paired up and placed two fish per mesh insert in a 10 L tank on the recirculating system, allowing pairs of fish to be tracked through the experiment. The location of each pair was randomly rotated daily to minimise tank location effects. Fish assigned to the HFD group were immersed in 5% chicken egg yolk for 2 hr daily over the course of 14 days (*Minchin and Rawls, 2017*). Control fish were placed in fresh system water for 2 hr daily. Both control and high-fat groups received an additional two feeds throughout the day.

## Image-based adipose morphology profiling in zebrafish

For each Nile Red-stained juvenile zebrafish, we took two images on a Leica M205 FCA fluorescence stereomicroscope with a 1× objective: a whole-animal image to allow us to measure total adiposity and SL, and a higher magnification image of the lateral subcutaneous adipose tissue along the zebrafish flank (hereafter referred to just as subcutaneous adipose or SAT). Images were always taken of the right side of the fish, as zebrafish adipose shows left-right asymmetry, with the right side containing the majority of pancreatic adipose. Images were first processed in Fiji/ImageJ and the 'Clear Outside' function was used to reduce image size and focus on the SAT area of interest or wider zebrafish. LD SAT images were either imported to Cellpose v2.2 and segmented using the 'cyto' model (*Stringer et al., 2021*), or segmented in Napari using the SAMCell plugin and the samcell-cyto-large pre-trained model. Automated segmentations were assessed and manually adjusted to correct errors. Segmented masks and object outlines were imported to Fiji/ImageJ, where xy coordinates for each segmented LD along with LD measurements were taken (including Feret's diameter for LD size). ImageJ macros used for the importation of Cellpose masks and measurement of xy coordinates are available at GitHub, copy archived at *Minchin, 2026*. Total adipose area and SAT area were segmented using Ilastik software as previously described (*Tandon et al., 2025*). SL was measured using the line tool in Fiji/ImageJ. Data files were coded prior to analysis such that investigators were blinded to experimental group assignment during quantification.

## Morphology value calculation

To quantify hyperplastic/hypertrophic adipose morphology, we followed methodology from *Arner et al., 2010*. Briefly, for a given total SAT area, we modelled the expected mean LD diameter using a generalised additive model (GAM):

$$\text{Feret} \sim s\left(\log\left(\text{AreaSum}\right)\right)$$

where Feret is the mean LD Feret diameter, AreaSum is the total SAT area, and s denotes a penalised thin-plate regression spline fitted using the mgcv package in R. The GAM was fitted, for each gene, to all control (Cas9-only injected) fish pooled across experiments. Morphology value was calculated at the individual-fish level as the deviation of the observed mean Feret diameter from the GAM-predicted value. Positive deviations indicate a hypertrophic phenotype (larger LDs than expected for

a given adipose depot size), while negative deviations indicate a hyperplastic phenotype (smaller, more numerous LDs).

## Statistical framework for multi-experiment comparisons

Genes were tested across multiple independent injection experiments (typically 2–3 experiments per gene, see *Figure 5*). Two genes (*aspa* and *lamb2*) had morphology data from only a single experiment and were analysed separately without batch correction. For genes with data from ≥2 experiments, we employed complementary statistical approaches to identify robust phenotypes while controlling for batch effects.

### KS tests

To assess distributional differences in morphology values between mutants and controls, we performed two-sample KS tests:

Pooled KS test: Control and mutant data were pooled across all experiments. A GAM was fit to pooled control data, and KS tests compared the distribution of control residuals to mutant deviations from the control GAM.

Stratified KS test: To validate that effects were consistent within experiments (rather than driven by batch differences), we performed KS tests within each experiment separately, fitting experiment-specific GAMs to control data. Within-experiment p-values were combined using Fisher's method:

$$\chi^2 = -2 \sum_{\{i=1\}}^{\{k\}} \ln(p_i)$$

where k is the number of experiments. The combined statistic follows a $\chi^2$ distribution with 2k degrees of freedom under the null hypothesis. Weighted mean effect sizes were calculated across experiments, weighting by sample size.

### Linear mixed models

To quantify effect sizes while accounting for batch variation, we fit LMMs with experiment as a random intercept using the lme4 package.

Intercept model (uniform shift):

$$\log(Feret) \sim \log(AreaSum) + Genotype + (1|Experiment)$$

This tests whether mutants have uniformly larger or smaller LDs across all adipose depot sizes.

Interaction model (size-dependent effect):

$$\log(Feret) \sim \log(AreaSum) \text{ x } Genotype + (1|Experiment)$$

This tests whether the allometric scaling relationship between LD size and total adiposity differs between genotypes. A significant interaction indicates that the mutant phenotype magnitude depends on total adipose tissue size.

Effect sizes from the intercept model were back-transformed to percentage changes in Feret diameter: % change = (exp(β) − 1)×100, where β is the genotype coefficient. 95% confidence intervals were calculated as (exp(β±1.96 × SE) − 1)×100.

### Multiple testing correction

Benjamini-Hochberg FDR correction was applied separately to: (i) pooled KS tests, (ii) stratified KS tests (Fisher's combined p-values), (iii) LMM intercept p-values, and (iv) LMM slope p-values. Adjusted p-values<0.05 were considered significant.

### Robustness criteria

Genes were classified as robust hits if they reached significance (adjusted p<0.05) in both the stratified KS test and LMM intercept test. This dual criterion ensures that: (i) the distributional effect is consistent within experiments (not a batch artifact), and (ii) the mean shift is statistically significant after accounting for repeated measures.

## Variance heterogeneity analysis

To test whether mutants showed altered phenotypic variance beyond mean shifts, we performed F-tests comparing the variance of morphology values between genotypes. For each gene, the variance ratio was calculated as $\sigma^2$(mutant)/$\sigma^2$(control). Higher-order distributional moments were calculated using the moments package in R: skewness (asymmetry of distribution) and excess kurtosis (tail weight relative to normal distribution). The percentage of observations exceeding 2 standard deviations from the mean ('tail percentage') was also calculated. p-Values from F-tests were corrected using Benjamini-Hochberg FDR.

## LMM robustness verification

To verify that LMM results were not artifacts of violated model assumptions, we performed sensitivity analyses using three alternative approaches:

Heteroscedastic LMM: Linear mixed models allowing different residual variances per genotype were fit using the nlme package, with variance structure specified as weights = varIdent (form = ~1|genotype).

Permutation test: Genotype labels were permuted 1000 times within each experiment to generate a null distribution of genotype effects. Permutation p-values were calculated as the proportion of permuted effects exceeding the observed effect.

Welch's t-test: Two-sample t-tests with Welch-Satterthwaite degrees of freedom correction (not assuming equal variance) were performed on morphology residuals.

Results were classified by consensus: 'ALL SIG' if all methods yielded p<0.05, 'ALL NS' if no methods were significant, or 'MIXED' if methods disagreed. Only genes with consistent significance across all sensitivity methods were considered to have robust LMM estimates.

## Single-experiment gene analysis

For *aspa* and *lamb2* (single experiment each for morphology), batch correction was not possible. These genes were analysed using standard linear regression:

$$\log\left(\text{Feret}\right) \sim \log\left(\text{AreaSum}\right) + \text{Genotype}$$

KS tests were also performed. Results are reported without FDR correction and should be interpreted as preliminary findings requiring validation.

## Software and code availability

All analyses were performed in R (version 4.5.1) using the following packages: mgcv for GAM fitting, lme4 and lmerTest for LMMs, nlme for heteroscedastic mixed models, and moments for distributional statistics. Code and data are available as a reusable pipeline at GitHub (*Minchin, 2026*).

## Materials availability

Zebrafish mutant lines *foxp1a*[ed116] and *foxp1b*[ed125] generated in this study are available from the corresponding author upon request.

# Acknowledgements

We thank Rosalyn Fong for help with initial zebrafish foxp1 genotyping strategies, and David Duneau, Will Cawthorn and Rob Semple for advice on modelling of adipose morphology. The work was funded by a British Heart Foundation PhD Studentship awarded to RW, and a BBSRC grant (BB/X009467/1) awarded to PT and JENM.

# Additional information

## Funding

| Funder | Grant reference number | Author |
|---|---|---|
| Biotechnology and Biological Sciences Research Council | BB/X009467/1 | Panna Tandon James Minchin |
| British Heart Foundation | REA PhD Studentship | Rebecca Wafer |

The funders had no role in study design, data collection and interpretation, or the decision to submit the work for publication.

## Author contributions

Rebecca Wafer, Data curation, Formal analysis, Validation, Investigation, Visualization, Methodology; Panna Tandon, Conceptualization, Resources, Data curation, Formal analysis, Supervision, Funding acquisition, Validation, Investigation, Visualization, Methodology, Writing – original draft, Project administration, Writing – review and editing; James Minchin, Conceptualization, Resources, Data curation, Software, Formal analysis, Supervision, Funding acquisition, Validation, Investigation, Visualization, Methodology, Writing – original draft, Project administration, Writing – review and editing

## Author ORCIDs

Rebecca Wafer ⓘ https://orcid.org/0009-0007-1139-3937
James Minchin ⓘ https://orcid.org/0000-0003-2316-1010

## Ethics

All zebrafish experiments were conducted in accordance with the UK Animals (Scientific Procedures) Act 1986 under project licence PP9112175. The protocols and project license were reviewed and approved by the University of Edinburgh Animal Welfare and Ethical Review Body.

Reviewer #1 (Public review): https://doi.org/10.7554/eLife.107327.3.sa1
Reviewer #2 (Public review): https://doi.org/10.7554/eLife.107327.3.sa2
Author response https://doi.org/10.7554/eLife.107327.3.sa3

# Additional files

## Supplementary files

MDAR checklist

Supplementary file 1. Differentially expressed genes from *Honecker et al., 2022*.

Supplementary file 2. The 102 candidate adipose morphology genes from this study.

Supplementary file 3. GO term enrichment analysis to prioritise candidate adipose morphology genes.

Supplementary file 4. Guide RNA sequences used for the F0 CRISPR screen.

## Data availability

Data, code and interactive HTML analysis files are available as a reusable pipeline at GitHub, copy archived at *Minchin, 2026*.

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
