## [Editor Report · eLife Assessment]

In this manuscript, Wafer and Tandon et al. present a thoughtful and well-designed genetic screen for regulators of adipose remodeling using zebrafish as a model system. This work is **valuable** because it uncovers several genes associated with adipose tissue hyperplastic hypertrophic morphology and diet-induced remodelingthe hat have considerable potential health impact. The rigorous phenotypic analyses and **compelling** evidence make this work a key resource for the field.

---

## [Referee Report · Reviewer #1 (Public review)]

In this manuscript, Wafer and Tandon et al. present a thoughtful and well-designed genetic screen for regulators of adipose remodeling using zebrafish as a model system. The authors cross-referenced several human adipocyte-related transcriptomic and genetic association datasets to identify candidate genes, which they then functionally tested in zebrafish. Importantly, the authors devised an unbiased microscopy-based screening platform to document quantitative adipose phenotypes with whole animal imaging, while also employing rigorous statistical methods. From their screen, the authors identified 3 genes that resulted in robust adipose phenotypes out of a total of 25 that were tested. Overall, this work will be an important resource for the field because of the genes identified from the screen, the quantitative screening pipeline, and the rigorous phenotypic analysis.

Comments on revisions:

The authors have far exceeded my expectations with their revised manuscript. All my questions and concerns from the original manuscript have been addressed by the authors. The additional data and analysis in Figure 6 and Supplementary Figure 8 are compelling and have greatly improved the manuscript.

---

## [Referee Report · Reviewer #2 (Public review)]

This manuscript by Wafer, Tandon et al., presents exciting new approaches for using the zebrafish CRISPR screening and imaging system to identify genes that are associated with hyperplastic and hypertrophic adipose morphology. This paper established valuable screening pipelines in zebrafish to identify genetic regulators that affect adipose tissue morphology by combining CRISPR with an imaging-based, comprehensive adipose spatial analysis platform. Starting from a human transcriptomic dataset with differentially expressed genes that separate small and large adipocytes, they eventually identified 3 genes that induce hyperplastic or hypertrophic phenotypes in zebrafish. From which, they focused on foxp1 gene, a transcription factor known to regulate tissue development. They discovered that the foxp1 mutant displays basal hypertrophic morphology and failed to undergo hypertrophic remodeling in response to a high-fat diet, suggesting a link between adipose tissue development and diet-induced remodeling response. Overall, this manuscript is extremely well-written, the data presented is quite compelling, and the identified novel genes that are associated with adipose tissue hyperplastic and hypertrophic morphology and diet-induced remodeling are very exciting.

Strength:

(1) Obesity remains a worldwide public health concern. The mechanisms underlying adipose tissue hypertrophic and hyperplastic adaptation remain unclear.

(2) This manuscript combined multiple omic datasets to identify candidate genes and performed a CRISPR-based screening to identify genes underlying adipose tissue development and adaptation. This new method will open opportunities that will facilitate our understanding and testing of new genetic mechanisms underlying the development of obesity.

(3) Using the screening approach, this paper successfully identified new genes that are associated with adipose tissue LD size change. More importantly, the paper provided further validation using a stable CRISPR line to show the phenotype in basal and HFD conditions.

(4) The experiments are extremely well-designed. Sample sizes are large. Statistical analysis is rigorous. Overall, this is a very high-quality study.

Author's response to the previous comments/weakness:

(1) In this revised manuscript, the authors provided new comprehensive spatial analyses of foxp1a and foxp1 b mutants in basal conditions as well as responding to high-fat feeding. The new data confirmed their initial findings and beautifully illustrated the spatiotemporal dynamics of the adipocytes in response to High-fat diet feeding.

(2) The authors have addressed all my comments, and I do not have further comments.

---

## [Author Response]

The following is the authors’ response to the original reviews.

Thank you for the thoughtful and constructive comments on our manuscript. We have carefully addressed all points raised, and believe the manuscript is substantially improved as a result. In particular, we have performed:

- Comprehensive spatial analysis of stable mutants. Following Recommendations for the authors comment #1, we performed spatial analysis by binning the anterior-posterior axis into 200 µm strata. This analysis validates our initial conclusions and reveals striking spatiotemporal dynamics, including profoundly blunted HFD responses in *foxp1b* mutants (68% reduction) and loss of spatial gradients in *foxp1a* mutants.

- Substantially enhanced the statistical rigour of the screen analysis. We have implemented stratified Kolmogorov-Smirnov tests (within-experiment testing, then combined via Fisher's method) alongside linear mixed models to control for batch effects. In the revised manuscript, we now focus on three hypertrophy genes – *foxp1b*, *txnipa* and *mmp14b* – which are robustly validated by both methods.

- Normalisation of adipose area to body size. To address concerns about developmental delay (Recommendations for the authors #2), we now normalise adipose area to standard length. With this normalisation, *foxp1b* single mutants show only a non-significant trend toward decreased adiposity (updated from our original analysis), while the hypertrophic LD morphology remains highly significant - demonstrating the phenotype is independent of body size and not a developmental delay.

- Revised title. As suggested by Recommendations for the authors comment #6, we have changed the title to: "A quantitative in vivo CRISPR-imaging platform identifies regulators of hyperplastic and hypertrophic adipose morphology in zebrafish"

- Extensive code and analysis availability. We now provide all code and extensive analysis pipelines in interactive HTML documents at https://github.com/jeminchin/zebrafish_adipose_morphology_screen

**Joint Public Review:**

We thank the reviewers for their thoughtful assessment of our work and their recognition of the rigorous experimental design, statistical approaches, and the utility of both the identified genes and screening pipeline for the field. We address their concerns below.

Weakness:Distinguishing developmental patterning from adipose tissue plasticity

We appreciate this important distinction and agree that separating developmental from adaptive effects is a key challenge in the field. We would like to make several points in response:

First, we acknowledge this limitation in our discussion and have now expanded this section to more explicitly address the interpretive boundaries of our approach. Our screening platform was intentionally designed to capture the outcome of genetic perturbation across development and early adaptation, as these processes are inherently intertwined during the establishment of adipose tissue.

Second, regarding the suggested analysis of lipid droplet size along the AP axis in response to HFD: we have now performed this analysis and include it as new Fig. 6 and new Supplemental Fig. 8 & 9. These data validate our initial conclusions and reveal striking spatiotemporal dynamics, including profoundly blunted HFD responses in *foxp1b* mutants (68% reduction) and loss of spatial gradients in *foxp1a* mutants. Further, these data provide additional resolution on regional responses to dietary challenge.

Third, we note that our stable mutant validation experiments (Figure 6) do begin to disentangle these effects by examining both baseline and HFD-challenged conditions in animals with constitutive genetic loss. However, we agree that definitive separation would require temporally controlled genetic manipulation, which we now acknowledge as an important future direction.

Lack of tissue-specific manipulations

We agree that tissue-specific approaches would strengthen mechanistic conclusions and have acknowledged this limitation in our revised discussion. The current study was designed as a discovery-focused screen to identify candidate regulators, with the understanding that mechanistic dissection would require follow-up studies employing tissue-specific tools.

We note that adipocyte-specific Cre/lox or Gal4-UAS approaches in zebrafish are feasible and represent an important next phase of investigation for the most promising candidates identified here, rather than a requirement for the current screening study. We have added text explicitly framing our findings as establishing genetic associations that warrant future tissue-autonomous investigation.

**Recommendations for the authors:**
(1) Analysis: In Figure 6, the authors state that foxp1b mutants "fail to undergo further hypertrophic remodeling in response to a high-fat diet (HFD)." Foxp1b mutant juveniles are already hypertrophic before the high-fat diet. After a high-fat diet, these mutants reach mean lipid droplet diameters similar to WT, approximately 65 µm, which the authors state earlier in the manuscript are "a potential upper limit of LD growth at this developmental stage." The authors should perform additional analysis of their existing data. Specifically, determine lipid droplet size by binning the AP axis as shown in Figure 3. The rationale is that lipid droplet size differences in response to HFD may be more evident when not considering the anterior populations of lipid droplets that have already reached maximum steady state size for this juvenile stage. This would not require any new experiments, just reanalyzing data similar to how they did in Figure 3.

We thank the reviewer for this excellent suggestion. We have performed the requested spatial analysis by binning the AP axis into 200 µm strata (Figure 3 approach). These data can be found in new Fig. 6H-M, and new Supplemental Figs 8 & 9. This new analysis verifies our initial conclusions, and also reveals several very interesting spatiotemporal dynamics

(i) Baseline hypertrophy in *foxp1b* mutants across AP strata

In support of our initial conclusion that *foxp1b* mutants have larger LDs at baseline, the spatial analysis confirms that on a control diet (baseline), *foxp1b* mutants have significantly larger LDs than WT across strata 1-5 (new Fig. 6I), ranging from +22.2 µm larger in strata 1 to +17.8 µm larger in strata 5 (all FDR-adjusted p < 0.05, linear mixed effects model). Extended analysis across all 15 strata is shown in Supplemental Figs. 8 & 9. By contrast, and also in support of our initial conclusion, foxp1a mutants showed no baseline hypertrophy on control diet (all strata p > 0.10, Supplemental Fig. 8).

(ii) *foxp1b* mutants show a profoundly blunted hypertrophic response to HFD

Using paired analysis (same fish on both control diet and after 14 days of high-fat diet) with a linear mixed effects model, we quantified the effect of HFD across all strata:

(A) Anterior/oldest strata (1-6): WT + HFD increases LD diameter by +25.1-28.1 µm (+52-58%, p < 0.0001). Whereas, *foxp1b* mutants + HFD only increase LD diameter by +7.5-11.7 µm (+12-19%, p < 0.003). Therefore, in the oldest/most anterior regions, containing the largest LDs, the hypertrophic response of *foxp1b* mutants to HFD is ~57% weaker than WTs.

(B) Posterior/newer strata (7-15): WT + HFD undergo significant increases in LD diameter of +17.7-23.7 µm (p < 0.024). However, in *foxp1b* mutants there is no significant hypertrophic response at all (p > 0.068), and hypertrophic effect sizes decline from +6.8 µm (stratum 7) to +0.4 µm (stratum 15).

(C) Overall effect: Averaged across all strata, WT + HFD LDs show +24.4 µm increase (p < 0.0001), whereas *foxp1b* mutant LDs only show a +7.7 µm increase with HFD (p = 0.020). Therefore, *foxp1b* mutants show a 68% reduction in hypertrophic growth in response to HFD compared to WT (Fig. 6K).

The consequence of these spatial dynamics is that WT SAT LDs - which start 22 µm smaller than *foxp1b* mutants on a control diet - undergo massive hypertrophy across all regions/strata in response to a HFD. Meanwhile, *foxp1b* mutants - starting larger than in WTs - show only a modest, spatially restricted response. This results in a convergence in LD size in early/anterior strata, but WT LDs actually surpass *foxp1b* mutant sizes in late/posterior strata (strata 14-15: +WT 14.7 µm larger on HFD, p = 0.028; Supplemental Figs. 8 & 9).

By contrast, *foxp1a* mutants retain the capacity for HFD-induced hypertrophy but show a ~35% weaker response than WT (p = 0.023) – significantly less severe than the 68% reduction in *foxp1b* mutants. Interestingly, *foxp1a* mutants after HFD show a reduction in the AP gradation of LD size observed in WT and *foxp1b* mutants (uniform +14.4 mm across all strata versus WT range of +26.4 mm anteriorly to +16.6 mm posteriorly), suggesting that *foxp1a* may regulate spatial heterogeneity in adaptive responses to HFD (Fig. 6L-M).

(iii) Developmental ceiling or impaired adaptive capacity?

The reviewer raises an important question about whether anterior adipose LDs have reached a "developmental ceiling." After conducting the spatial analysis suggested by the Reviewer, we now believe several lines of evidence support an intrinsic defect in HFD-induced hypertrophy in *foxp1b* mutants, rather than reaching a developmentally determined limit:

First, *foxp1b* mutants show reduced responses across ALL strata, not just anterior regions. The attenuation extends throughout the entire AP axis (57% reduction in strata 1-6, complete loss of response in strata 7-15). If anterior adipocytes had simply reached a size ceiling, we would expect normal responses in posterior regions where cells are smaller - but we don't observe this.

Second, in posterior/newer regions of SAT (strata 14-15) the hypertrophic response to HFD in *foxp1b* is so limited that WT LDs actually become larger than *foxp1b* mutant LDs (+14.7 mm larger, p = 0.028; Supplemental Fig. 9). This demonstrates that these LD sizes are not developmentally limiting and argues for intrinsic hypertrophic defects in response to HFD.

Third, *foxp1a* mutants provide an important control. These mutants show no baseline hypertrophy (all strata p > 0.10) yet still exhibit blunted hypertrophic responses to HFD (~35% reduction, p = 0.023), proving that reduced HFD responses can occur independently of baseline hypertrophy.

We have updated the Results and Discussion to reflect these new conclusions. Methods have been updated to include the spatial analysis approach.

(2) Adipose morphogenesis in WT is a function of standard length, as shown by the authors. At juvenile stages, foxp1 mutants are both smaller and have reduced adipocyte coverage, while adults show normal body length and very subtle adipose phenotypes. Can the authors demonstrate that the observed defects in foxp1 mutant juveniles are bona fide phenotypes rather than a developmental delay?

We thank the reviewer for this key point. We agree it is critical to distinguish true *foxp1b*-dependent phenotypes from potential developmental delay. Importantly, our data strongly argue against a simple developmental delay. We show that LD size scales with body size in Fig. 3G, with smaller zebrafish having smaller LDs and larger zebrafish having larger LDs. In contrast to a developmental delay, our data show that *foxp1b* single and *foxp1a;foxp1b* double mutants are smaller (reduced standard length) but have larger LDs (Fig. 6E,G). This dissociation between body size and LD size is the opposite of what would be expected from developmental delay.

To account for the body size difference, we have now normalised adipose area to standard length (Fig. 6F). With this normalisation, *foxp1b* single mutants show only a non-significant trend toward decreased adiposity, whereas *foxp1a;foxp1b* double mutants remain significantly reduced. This represents a change from our original analysis and we have updated the text accordingly. Critically, despite normalised adipose area showing only a trend in *foxp1b* singles, the hypertrophic LD morphology remains highly significant (Fig. 6G), demonstrating that the morphological phenotype is robust and independent of overall body size.

We have clarified this interpretation in the Results and Discussion.

(3) What was the rationale for selecting one amongst paralogous genes for the screen? For example, why did the authors choose ptenb rather than ptena?(4) Point 3 is particularly relevant for the final six genes that resulted in adipose phenotypes. Why did the authors choose not to target both paralogs, given that multi-plexed F0 CRISPR targeting is feasible in zebrafish (PMID: 29974860).

We answer Points 3 & 4 together here.

We used the DIOPT (DRSC Integrative Ortholog Prediction Tool) orthology tool to identify the zebrafish paralogue with the highest orthology score to each human gene. This tool integrates predictions from 20 orthology databases to generate a composite score. We selected the paralogue with the highest DIOPT score for each gene. For example, we selected *ptenb* over *ptena* because it showed a higher predicted orthology to human PTEN.

We acknowledge this approach has important limitations, including orthology scores not necessarily predicting functional equivalence (ie, the "most orthologous" paralogue may not be the one with the most relevant adipose tissue function in zebrafish). We acknowledge that this may mean we have missed genuine hits - testing only one paralogue means we could fail to identify genes where the "less orthologous" paralogue has the relevant adipose function.

Our findings with Foxp1 paralogues both validate this approach and reveal its limitations. The higher-scoring paralogue *foxp1b* (DIOPT score = 13/19) showed the more severe phenotype, validating our prioritisation. However, the lower-scoring paralogue *foxp1a* (DIOPT score = 5/19), which we tested subsequently, showed a distinct but significant phenotype (altered spatial patterning) – a finding that would have been missed had we not pursued secondary validation.

For future screens where comprehensive hit identification is the goal, multiplexed targeting of all paralogues would be valuable, though this may complicate interpretation of paralogue-specific phenotypes. We have discussed this in the Discussion.

(5) General framework and limitations: The analysis platform presented in the manuscript cannot separate the developmental effects from adipose tissue plasticity/remodeling. Potential approaches that may help address this concern include: (a) establishing a baseline model to illustrate how WT fish respond to high-fat diet (HFD); (b) showing how mutants with hyperplasticity (opposite effects of foxp1 mutants) respond to HFD; (c) examining whether foxp1 gene expression level changes in response to HFD. However, these approaches (especially a and b) would require extensive experimental work and may be beyond the scope of this study. Without further evidence or data support of adipose tissue plasticity and remodeling, the author may want to emphasize in the background and discussion sections how adipose tissue development may affect plasticity and adaptation, and soften the tone of how genes may directly regulate adipose tissue plasticity and adaptation.

We thank the reviewer for this comment about the relationship between adipose development and plasticity/remodelling. We agree this is an important issue as we are looking in juvenile fish that are still growing. Therefore, when we feed them HFD and see LDs get bigger – is this diet-induced remodelling or just accelerated normal development (ie, growth that would happen anyway, but occurring faster due to more nutrients)?

To address the reviewer's specific suggestions:

(A) Baseline model of WT HFD response: We have now performed detailed spatial analysis of WT responses to HFD (new Fig 6H-M, Supplemental Figs. 8 & 9). This analysis establishes a comprehensive baseline for hypertrophic responses to HFD in developing adipose tissue. In summary, WT fish show robust, statistically significant and spatially-graded hypertrophic responses to HFD across the entire AP axis, with responses ranging from +28.1 mm anteriorly to +17.7 mm posteriorly.

We agree with the Reviewer that separating developmental from adaptive processes in growing juvenile fish is challenging. Importantly, we believe *foxp1a* mutants provide compelling genetic evidence that we are studying adaptive responses rather than purely developmental processes. *foxp1a* mutants have normal baseline LD sizes on control diet (demonstrating *foxp1a* is not required for developmental adipose expansion), yet when challenged with HFD show significantly reduced hypertrophic expansion and reduction of spatial gradient. This genetic dissociation strongly argues we are observing adaptive capacity rather than developmental growth rate.

(B) Hyperplastic mutants:

We agree that analysis of hyperplastic mutants would provide valuable complementary information about tissue remodelling capacity. However, as the reviewer anticipated, this would require: (1) generating stable lines of the appropriate hyperplastic mutants, (2) conducting paired HFD feeding studies, (3) performing spatial morphometric analysis comparable to our *foxp1* studies, and (4) potentially distinguishing hyperplastic vs hypertrophic contributions to expansion. We agree this constitutes substantial additional experimental work beyond the scope of the current manuscript, though it represents an important direction for future studies.

(C) *foxp1* expression changes in HFD:

Unfortunately, we do not have SAT samples from HFD-treated fish preserved for RNA analysis, and therefore cannot assess whether *foxp1* expression levels change in response to dietary challenge. This would be valuable for future studies to determine whether *foxp1* genes are dynamically regulated during metabolic adaptation or function as constitutive regulators of adaptive capacity.

Following the Reviewer's guidance, we have revised throughout the manuscript to more carefully distinguish developmental patterning from metabolic adaptation.

(6) Title: In the absence of experimental results that can distinguish between developmental effects from adipose tissue plasticity/remodeling, such as those mentioned above, the manuscript title is not accurate and should therefore be revised to be something like "hyperplastic and hypertrophic adipose morphology."

We have now altered the title as the Reviewer suggested to “A quantitative in vivo CRISPR-imaging platform identifies regulators of hyperplastic and hypertrophic adipose morphology in zebrafish”

Minor:(7) In mice studies, deleting foxp1b in adipose tissue protects mice from diet-induced obesity, while overexpressing foxp1b in adipose tissue promotes diet-induced obesity (Liu et al., Nature Communication, 2019). These overall phenotypes and foxp1b-mediated effects appear to be contradictory to what is observed in the zebrafish model. Can the authors also provide more evidence/discussion on why such a difference occurs comparing zebrafish and mice models?

We thank the reviewer for this important comparison. We believe the apparent contradictions reflect (1) differences in adipose tissue thermogenic capacity - between species possibly, but also between functionally distinct depots and (2) whole-organism versus tissue-specific experimental approaches.

(1) Different adipose tissue biology: browning-prone vs browning-resistant adipose

Liu et al. (2019, PMID: 31699980) demonstrated that adipose-specific deletion of Foxp1 in mice increases thermogenesis and browning of SAT, with protection from diet-induced obesity (DIO) and improved insulin sensitivity. Conversely, Foxp1 overexpression impaired adaptive thermogenesis and promoted DIO. Mechanistically, Foxp1 directly represses β3-adrenergic receptor transcription, thereby inhibiting the thermogenic program. Strikingly, mouse Foxp1-deleted adipocytes displayed smaller, multilocular lipid droplets characteristic of brown/beige adipocytes.

These morphological outcomes initially appear opposite to our zebrafish findings: mouse Foxp1 mutants have smaller adipocytes (due to browning), while zebrafish *foxp1b* mutants have larger lipid droplets (hypertrophy). We believe this fundamental difference may reflect the propensity of adipose tissue to undergo adaptive thermogenesis.

While it was recently discovered that zebrafish possess thermogenic epicardial adipose tissue (PMID: 38507414), in general zebrafish adipose is not considered thermogenic, and zebrafish as ectotherms are thought to lack adaptive thermogenesis for thermoregulation. The exact thermogenic potential of zebrafish adipose remains to be fully characterised, but potential differences in thermogenic capacity between mouse and zebrafish adipose may help explain the distinct phenotypic outcomes.

Importantly, Liu et al. studied mouse inguinal subcutaneous WAT - the depot most prone to browning in rodents. It remains unclear what role Foxp1 plays in browning-resistant mammalian WAT depots, where thermogenic conversion does not readily occur. In such depots, Foxp1 loss might produce phenotypes more similar to our zebrafish findings - dysregulated white adipose function without browning.

The above hypothesis suggest that browning responses may mask other roles for Foxp1 in WAT. Interestingly, although not quantified in the paper, Liu et al.’s Foxp1 overexpression model (Ap2-Foxp1) appeared to reduce adipocyte size despite suppressing Ucp1 expression and reducing lipolysis. These data suggest more complex roles and indicate that Foxp1’s control of adipocyte size might extend beyond simply regulating thermogenesis and may involve coordinating the balance between hyperplastic versus hypertrophic expansion.

Furthermore, human subcutaneous WAT is not as prone to browning as mouse inguinal WAT. Human browning occurs primarily in specialised depots (e.g. supraclavicular, deep neck), while the majority of human adipose tissue represents constitutive white adipose with limited thermogenic capacity. Therefore, it remains an open question whether FOXP1's primary physiological role in humans relates to thermogenesis regulation (in specialised depots) or white adipose metabolic control (in the majority of adipose tissue). Zebrafish findings examining constitutive WAT function (admittedly the lack of adaptive thermogenesis in zebrafish is presumed at this stage) may be more relevant to human adipose than initially appear.

(2) Whole-organism vs tissue-specific effects on metabolic health

A second apparent contradiction concerns metabolic outcomes: mouse adipose-specific Foxp1 deletion improves metabolic health (Liu et al.), whereas our zebrafish whole-organism *foxp1b* mutants display metabolic dysfunction (baseline hypertrophy, impaired HFD response, hyperglycaemia and fatty liver). We believe this discrepancy reflects comparison of whole-animal mutants (zebrafish) to tissue-specific deletions (mouse), rather than opposite adipose tissue functions.

Critically, Foxp1 has established roles in hepatic glucose metabolism. Zou et al. (PMID: 26504089) demonstrated that hepatic Foxp1 inhibits expression of gluconeogenesis genes and decreases hepatic glucose production and fasting blood glucose by competing with Foxo1 for binding of insulin responsive gluconeogenic genes. In line with these observations, we observe fatty liver and hyperglycaemia in *foxp1a;foxp1b* double mutant zebrafish (data not shown), suggesting that the metabolic dysfunction in our whole-animal mutants may be driven primarily by hepatic Foxp1 loss rather than adipose-specific effects.

We have expanded on the points raised here in the Discussion.

(8) Line 522-524: "The major phenotype in foxp1a mutants was impaired adipose expansion following HFD, suggesting failure to respond to diet-induced stress signals". In the presented Figure 6j, foxp1a mutant expands adipose LD size following HFD, similar to the control, which is contradictory to the statement above. Please clarify.

We thank the reviewer for highlighting this apparent inconsistency and apologise for imprecise wording. These measurements are actually consistent but refer to different scales of analysis.

Tissue level (Supplementary Fig. 7): *foxp1a* mutants show significantly reduced total adipose expansion (based on whole-animal Nile Red images) compared to wild-type fish on HFD—this is what we refer to as "impaired adipose expansion."

Cellular level (Fig. 6L-M): At the individual adipocyte level, *foxp1a* mutants show statistically significant increases in LD diameter following HFD. However, the magnitude is reduced by ~35% compared to wild-type (mutants: +14.4 µm; WT: +22.2 µm; p = 0.023).

We have revised the text to more precisely state "reduced adipose expansion" rather than "impaired expansion" to avoid implying complete failure to respond.